# A hypothalamic circuit for circadian regulation of corticosterone secretion

Oscar D. Ramirez-Plascencia [1,2,5], Roberto De Luca [1,2,5], Natalia L. S. Machado [1,2], Dominique Eghlidi[1,2], Mudasir A. Khanday[1,2], Sathyajit S. Bandaru[1,2], Francesca Raffin [1,2,3], Nina Vujovic[2,4], Elda Arrigoni[1,2,6] & Clifford B. Saper [1,2,6] ✉

There is a strong circadian rhythm of corticosteroid secretion in animals and humans, but the circuit for translating the suprachiasmatic (SCN) clock light-dark cycle rhythm into an increase in corticosteroid secretion beginning several hours before the active period is not known. We show here that in male mice, this rhythm depends upon input from the SCN to the subparaventricular zone (SPZ), and then to the dorsomedial nucleus of the hypothalamus (DMH). Both glutamatergic and GABAergic DMH neurons are required for the daily surge of corticosteroid secretion in anticipation of the active period. Glutamatergic DMH neurons directly excite paraventricular nucleus corticotrophin-releasing hormone (PVH-CRH) neurons, whereas DMH GABA neurons disinhibit PVH-CRH neurons via a relay in GABAergic neurons in the caudoventral PVH. This circuit underlies the daily surge in corticosteroid secretion that is temporally linked but phase advanced compared to the SCN activity cycle.

PVH[CRH] neurons play a key role in the secretion of corticosteroids by releasing CRH into the hypothalamic-hypophysial portal circulation to stimulate the anterior pituitary release of adrenocorticotropic hormone (ACTH). ACTH then stimulates the adrenal cortex to secrete cortisol in humans and corticosterone (Cort) in rodents. The secretion of Cort follows a daily rhythm, with peak levels typically occurring just before the onset of the active phase, thereby regulating behavior, metabolism, and immune response. Trough levels occur toward the onset of the sleep phase, reflecting the natural dip in hormone secretion during the body's resting period[1,2]. This rhythmic secretion pattern reflects the intricate coordination of various neural and hormonal pathways involved in circadian regulation. Loss of the circadian rhythm of Cort secretion in rats after lesions of the suprachiasmatic nucleus (SCN) provided some of the first evidence for the role of the SCN as the brain's master biological clock[3]. However, the mechanism by which the SCN influences Cort secretion remains unknown. The SCN sends only limited axons to the PVH, where they mainly terminate in subregions of the PVH that control autonomic activity[4,5] and spare

the PVH[CRH] neurons. Indeed, the bulk of SCN efferents terminate in an arc stretching caudally and dorsally from the SCN, encompassing the subparaventricular zone (SPZ) and rostral dorsomedial nucleus of the hypothalamus (DMH)[6,7]. SPZ neurons are predominantly GABAergic[8–10], and lesion studies have shown that the ventral SPZ participates in the circadian regulation of the sleep-wake cycle, locomotor activity (LMA) and Cort release, while the dorsal part is involved in rhythms of body temperature (Tb) regulation[11]. Downstream, non-specific ablation of DMH neurons in rats eliminates the daily peak in Cort secretion, reducing the overall daily levels[12]. Stimulation of the DMH increases ACTH and Cort release[13,14]. These findings led us to hypothesize that the daily peak in Cort could be regulated by the SCN through successive relays in the SPZ and the DMH, which, in turn, stimulates the PVH[CRH] neurons to trigger Cort release[15].

To test this model, first we evaluated the role of SPZ[Vgat] neurons in transmitting the temporal information from the SCN to regulate the daily Cort release by selectively ablating them or preventing their GABA release. Then, as the DMH is a heterogeneous region containing

[1]Department of Neurology, Beth Israel Deaconess Medical Center, Boston, MA, USA. [2]Division of Sleep Medicine, Harvard Medical School, Boston, MA, USA. [3]Department of Biology and Biotechnology "Lazzaro Spallanzani", University of Pavia, Pavia, PV, Italy. [4]Departments of Medicine and Neurology, Brigham and Women's Hospital, Boston, MA, USA. [5]These authors contributed equally: Oscar D. Ramirez-Plascencia, Roberto De Luca. [6]These authors jointly supervised this work: Elda Arrigoni, Clifford B. Saper. ✉e-mail: csaper@bidmc.harvard.edu

roughly equal proportions of glutamatergic and GABAergic neurons, we examined the roles of both of these DMH cell types in the circadian regulation of Cort levels. We tested whether the DMH[Vglut2] neurons were necessary for the circadian regulation of Cort by either ablating or inhibiting them or deleting the *Vglut2* gene in the DMH, preventing glutamate release. We then tested the effect of chemogenetic activation of DMH[Vglut2] neurons on Cort levels and used Channelrhodopsin-assisted circuit mapping (CRACM) to determine the synaptic effects of DMH[Vglut2] → PVH[CRH] input. We also assessed the effect of the activation, as well as ablation or inhibition of the DMH[Vgat] neurons, and whether deleting the *Vgat* gene in the DMH affected the daily rhythm of Cort release. Finally, we examined the role of the caudal ventral part of the PVH and adjacent peri-PVH area GABAergic (cvPVH[Vgat]) neurons in disinhibiting the PVH[CRH] neurons through a polysynaptic circuit (DMH[Vgat] → cvPVH[Vgat] → PVH[CRH]).

## Results

### SPZ[Vgat] neurons, but not GABA release, are necessary for the circadian regulation of Cort

To evaluate whether the SPZ GABAergic neurons participate in regulating circadian Cort release, we placed injections in the SPZ of *Vgat-ires-Cre::L10* mice of a viral vector that constitutively expresses mCherry in neurons lacking Cre-recombinase (Cre), but in cells that express Cre instead produces diphtheria toxin A (DTA), which induces cell death (AAV10-hSyn-mCherry-DIO-DTA). We placed injections of an AAV-EGFP virus (*AAV8-DIO-EGFP*) in the same site in control mice. After four weeks to allow the full expression of DTA or EGFP, we measured Cort levels in blood from DTA injected and control mice in a 12:12 light:dark (LD) cycle, by tail nicks at four time points (ZT 1, 7, 13, 19, taken in random order with at least 30 hrs between samplings), and at the same time points in constant darkness (DD) (CT 1, 7, 13, 19), starting on the third day in DD at CT13, and then every 30 h. Simultaneously, we recorded the circadian rhythms of locomotor activity (LMA) and body

temperature (Tb) for 15 days in LD, and then 15 days under DD. From a total of 12 mice injected with DTA, 8 mice were identified by a blinded observer with bilateral DTA injection covering at least 70% of the SPZ; in mice that also expressed L10-GFP in the Vgat neurons, there was approximately 75% cell loss in the SPZ (Supplementary Fig. 1). Because some injections encroached on the borders of the SCN, we also did cell counts within the SCN in these mice and found less than 15% cell loss (Fig. 1a, b and Supplementary Fig. 1a). The mice with injections that missed the target were considered anatomical controls. The ablation of the SPZ[Vgat] neurons reduced the Circadian Index (CI, comparing the levels during the light vs the dark period; for method of calculation of CI see Statistical Analysis in the Materials and Methods) of Cort secretion compared with the control animals by $66.6 \pm 9.1\%$ in LD ($p = 0.008$) and $97.7 \pm 5.9\%$ in DD ($p < 0.001$). The peak Cort level was reduced from control levels of $23.4 \pm 1.7$ ng/ml in LD and $28.1 \pm 3.9$ ng/ml in DD, to $12.8 \pm 3.0$ ng/ml in LD and $4.8 \pm 1.1$ ng/ml in DD in SPZ[Vgat] ablated mice; (Fig. 1c–e). The SPZ[Vgat] ablations also caused a dramatic reduction in the CI of locomotor activity (LMA) by $51.1 \pm 3.5\%$ in LD ($p < 0.001$) and by $79.7 \pm 4.8\%$ in DD ($p < 0.001$), mostly by reducing the general LMA during the (presumptive) dark period (Supplementary Fig. 1b–i). The CI of body temperature (Tb) was reduced by $50.42 \pm 5.6\%$ in LD ($p < 0.001$) and $58.34 \pm 6.5\%$ in DD ($p < 0.001$; Supplementary Fig. 1j–q). The anatomical controls (missed injections) did not show any changes in LMA or Tb.

To evaluate whether the temporal information was transmitted by GABA release from the SPZ[Vgat] neurons, we placed injections in the SPZ of *AAV8-SYN-EGFP-iCre* (*AAV8-EGFP-iCre*) or a control virus (*AAV8-DIO-EGFP*) in *Vgat[loxP/loxP]* (*Vgat-flox*) mice to delete expression of functional Vgat protein and so to abolish GABA transmission from the transfected neurons. From 12 injected mice, 7 received a bilateral injection of *AAV8-EGFP-iCre* covering at least 70% of the SPZ without extending into the SCN (Fig. 1f, g and Supplemenary Fig. 2a). Surprisingly, deletion of Vgat from the SPZ neurons did not affect the circadian release

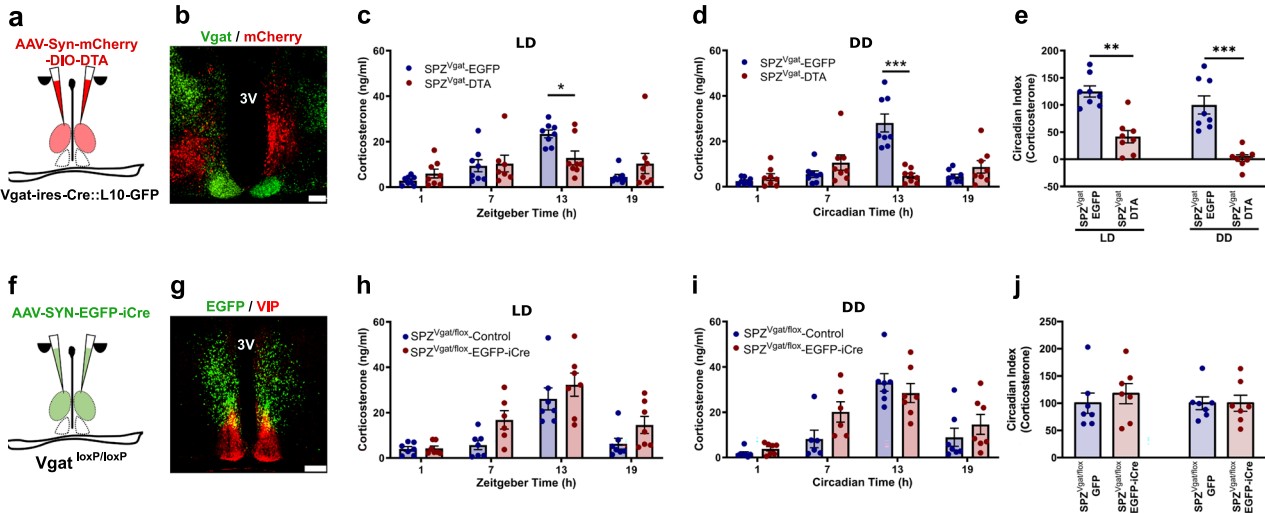

**Fig. 1 | SPZ[Vgat] neurons, but not GABA release, are necessary for maintaining the circadian rhythm of Cort secretion. a** Schematic drawing of SPZ[Vgat] neuron ablation by DTA. **b** Representative image of a section from an animal with SPZ[Vgat] neuron ablation, showing GABAergic neurons (in green, native signal) and the injection site in red (non- SPZ[Vgat] neurons expressing mCherry, native signal). Notice the remaining green GABAergic neurons in the SCN, and the almost complete elimination of GABAergic neurons just dorsally in the SPZ. Red mCherry was expressed by non-GABAergic cells in the injection site. Representative of 8 cases. **c** Ablation of SPZ[Vgat] neurons prevented the daily Cort increase at ZT13 in LD (*Two-way ANOVA*; *Tukey's* multiple comparisons test. ZT13 SPZ[Vgat]-EGFP vs SPZ[Vgat]-DTA: *$p = 0.025$, $n = 8$) and (**d**) at CT13 in DD (*Two-way ANOVA*; *Tukey's* multiple comparisons test. CT13 SPZ[Vgat]-EGFP vs SPZ[Vgat]-DTA: ***$p < 0.0001$, $n = 8$). **e** The circadian

index (CI, for method of calculation of CI, see Statistical Analysis in the Materials and Methods) of Cort secretion was reduced by $66.6 \pm 9.1\%$ in LD (*Unpaired t-test, Two-tailed: $t = 5.434$, df = 14, **$p = 0.002$, $n = 8$*) and by $97.7 \pm 5.9\%$ in DD (*Unpaired t-test, Two-tailed: $t = 5.534$, df = 14, ***$p < 0.0001$*). **f** Schematic representation of *Vgat* gene deletion in the SPZ. **g** Representative micrograph showing EGFP expression in the SPZ neurons in which the *Vgat* gene was deleted (green), just dorsal to the SCN[VIP] neurons (shown immunohistochemically in red). Representative of 7 cases. **h** Mice with *Vgat* gene deletion in the SPZ showed no change in the daily Cort peak either in LD or (**i**) DD photoperiod ($n = 7$). **j** No changes were detected in the Cort CI ($n = 7$). LD, Light:Dark photoperiod; DD, Constant darkness, Reference scale bar = 200 μm; 3 V, third ventricle. Data are presented as mean ± SEM.

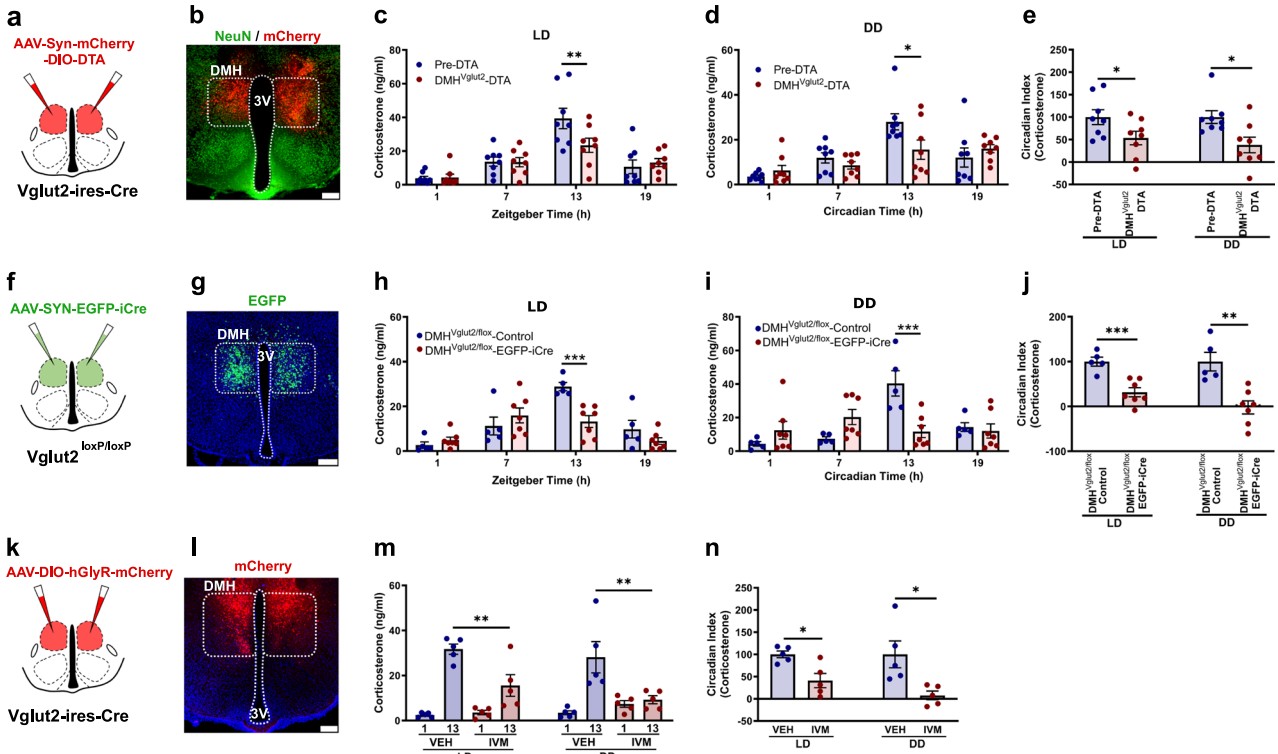

**Fig. 2 | DMH^Vglut2 neurons are necessary to maintain the endogenous circadian rhythm of Cort release. a** Schematic of the DMH^Vglut2 neuron ablation by DTA (**b**) Representative image of a section from an animal with DMH^Vglut2 neuron ablation, showing the injection site in red (non-DMH^Vglut2 neurons expressing mCherry) and green NeuN staining as a reference marker. Representative of 8 cases. **c** Ablation of DMH^Vglut2 neurons reduced the circadian peak of Cort at ZT13 in LD (*Two-way ANOVA*; *Tukey's* multiple comparisons test. ZT13 Pre-DTA vs DMH^Vglut2-DTA: **$p = 0.007$, $n = 8$) and (**d**) in DD at CT13 (*Two-way ANOVA*; *Tukey's* multiple comparisons test. CT13 Pre-DTA vs DMH^Vglut2-DTA: *$p = 0.013$, $n = 8$). **e** The Cort circadian index (CI) in the DMH^Vglut2 ablated mice was reduced by $46.6 \pm 14.2\%$ in LD (*Paired t-test, Two-tailed*: $t = 2.225$, $df = 14$, *$p = 0.043$, $n = 8$) and $62.0 \pm 17.3\%$ in DD (*Paired t-test, Two-tailed*: $t = 2.775$, $df = 14$, *$p = 0.014$, $n = 8$). **f** Schematic representation of *Vglut2* gene deletion in the DMH. **g** Representative micrograph showing EGFP expression in the DMH neurons in which the *Vglut2* gene has been deleted (in green). Representative of 7 cases. **h** *Vglut2* deletion in DMH diminished the Cort peak at ZT13 in LD (*Two-way ANOVA*; *Tukey's* multiple comparisons test. ZT13 DMH^Vglut2/flox-Control [$n = 5$] vs DMH^Vglut2/flox-EGFP-iCre [$n = 7$]: ***$p = 0.0007$) and entirely prevented the peak at CT13 in DD (**i**) *Two-way ANOVA*; *Tukey's* multiple

comparisons test. CT13 DMH^Vglut2/flox-Control [$n = 5$] vs DMH^Vglut2flox-EGFP-iCre [$n = 7$]: ***$p = 0.0002$). **j** The Cort CI was reduced by $68.2 \pm 10.1\%$ in LD (*Unpaired t-test, Two-tailed*: $t = 4.621$, $df = 10$, ***$p = 0.0009$, $n = 7$) and by $102.1 \pm 14.6\%$ in DD (*Unpaired t-test, Two-tailed*: $t = 4.167$, $df = 10$, **$p = 0.0019$, $n = 7$). **k** Schematic of the chemogenetic inhibition of DMH^Vglut2 neurons expressing hGlyR by IVM administration. **l** Representative micrograph of hGlyR-mCherry expression in the DMH (in red). Representative of 5 cases. **m** IVM administration diminished the Cort rise at ZT13 in LD (*Two-way ANOVA*; *Tukey's* multiple comparisons test. ZT13 VEH vs IVM: **$p = 0.006$, $n = 5$) and almost entirely prevented it at CT13 in DD compared with vehicle (*Two-way ANOVA*; *Tukey's* multiple comparisons test. CT13 VEH vs IVM: **$p = 0.001$). **n** The Cort CI was reduced one day after the administration of IVM by $58.9 \pm 15.9\%$ in LD (*Paired t-test, Two-tailed*: $t = 3.334$, $df = 8$, *$p = 0.01$, $n = 5$, mean and $\pm$ SEM) and by $92.3 \pm 9.9\%$ in DD (*Paired t-test, Two-tailed*: $t = 2.903$, $df = 8$, *$p = 0.019$, $n = 5$). In all cases we visualize the native signal except for the hGlyR signal that was enhanced with immunofluorescence for mCherry. LD Light:Dark photoperiod, DD Constant dark; Reference scale bar = 200 μm; 3 V, third ventricle. Data are presented as mean ± SEM.

---

of Cort (Fig. 1h–j). By contrast, the circadian rhythms of LMA and Tb were reduced by the deletion of *Vgat* from SPZ^Vgat cells, although to a lesser degree than after ablation of those neurons. For instance, the CI of LMA was reduced by $39.75 \pm 8.9\%$ in DD ($p < 0.001$; Supplementary Fig. 2b–i), and the CI of Tb was reduced by $25.7 \pm 4.6\%$ in LD ($p = 0.006$) and $38.6 \pm 7.3\%$ in DD ($p < 0.001$; Supplementary Fig. 2j–q). Taken together, these data indicate that the SPZ^Vgat neurons are necessary to transmit temporal information for circadian regulation of Cort, LMA and Tb from the SCN, but that the release of GABA alone cannot account for their influence on circadian regulation.

**Glutamate signaling by DMH^Vglut2 neurons is necessary for the increase in Cort at the beginning of the active period**

DMH neurons are downstream of the SPZ and have been shown in rats to be necessary for circadian rhythms of Cort. As DMH ablations cause a reduction in overall Cort secretion, we hypothesized that circadian Cort secretion may be under the control of excitatory glutamatergic DMH neurons. We therefore next examined the role of the DMH^Vglut2 neurons in the circadian regulation of Cort. We first measured Cort

levels in blood from 14 *Vglut2-ires-Cre* mice in a 12:12 light:dark (LD) cycle, at the same four time points (ZT 1, 7, 13, 19), and at the same time points in constant darkness (DD) (CT 1, 7, 13, 19). Simultaneously, we recorded the baseline circadian rhythms of LMA and Tb for 12 days in LD, and then 12 days under DD. We then placed injections into the DMH of *AAV10-hSyn-mCherry-DIO-DTA*, and after four weeks to allow full expression of DTA, we measured Cort levels in LD and DD as described, and recorded LMA and Tb in the mice for 12 days in LD and 12 days in DD. From a total of 14 mice, we identified 8 mice in which the injection covered at least 70% of the DMH bilaterally (Fig. 2a, b and Supplementary Fig. 3a) and these mice were used for further analysis. DMH^Vglut2 ablation reduced the Cort peak at the beginning of the active phase (ZT 13) from $39.4 \pm 6$ ng/ml in LD and $27.9 \pm 3.5$ ng/ml in DD before *AAV10-hSyn-mCherry-DIO-DTA* injections, to $23.4 \pm 4.2$ ng/ml in LD ($p = 0.007$, Fig. 2c) and $15.5 \pm 4.3$ ng/ml in DD after DMH^Vglut2 neuron ablation ($p = 0.013$, Fig. 2d). The CI of Cort secretion was reduced by $46.6 \pm 14.2\%$ in LD ($p = 0.043$) and $62.0 \pm 17.3\%$ in DD after the ablation ($p = 0.014$, Fig. 2e). To determine the specificity of the DMH^Vglut2 ablations for circadian increases in Cort, we measured the Cort response to

stress of the mice with the ablation by subjecting them to 1 h of restraint at ZT3. However, we did not detect changes in the Cort levels induced by restraint stress, indicating that there was no impairment in overall Cort secretion and that DMH[Vglut2] neurons do not play a role in restraint stress-induced Cort secretion (Supplementary Fig. 3b). The ablation of DMH[Vglut2] neurons also reduced the peak in LMA at the transitions between active and inactive periods both in LD and DD. Because these transition periods spanned the light and dark cycles, there was no change in the CI of LMA, although the amplitude of the LMA rhythm in DD was reduced in cosinor analysis (Supplementary Fig. 3c–j). In contrast, ablation of DMH[Vglut2] neurons reduced Tb during the light phase in LD and in both presumptive light and dark phases in DD. As a result, the CI and cosinor amplitude of the Tb rhythm was increased during LD but not DD (Supplementary Fig. 3k–r).

We then evaluated whether the effect of ablation of DMH[Vglut2] neurons on Cort rhythms was due to the loss of glutamate transmission, as opposed to other possible transmitters expressed in DMH[Vglut2] neurons. We therefore injected the DMH of *Vglut2[loxP/loxP]* (*Vglut2-flox*) mice with either *AAV8-EGFP-iCre* or a control virus *AAV8-DIO-EGFP* and recorded LMA, Tb and Cort levels in both LD and DD. Seven of the 11 mice had injections covering at least 70% of the DMH bilaterally and were compared with 5 mice with *AAV8-DIO-EGFP* injections in the DMH of *Vglut2-flox* mice as controls (Fig. 2f, g, Supplementary Fig. 4a). In these animals, the Cort levels at ZT13 were reduced in LD from 28.9 ± 1.9 ng/ml to 13.2 ± 2.6 ng/ml ($p < 0.001$, Fig. 2h) and in DD from 40.4 ± 7.5 ng/ml to 11.7 ± 3.5 ng/ml in DMH[Vglut2/flox]-Control and DMH[Vglut2/flox]-EGFP-iCre mice, respectively ($p < 0.001$, Fig. 2i). The CI of Cort was reduced by 68.2 ± 10.1% in LD ($p < 0.001$) and by 102.1 ± 14.6% in DD ($p = 0.001$, Fig. 2j). Both the LMA and Tb were reduced during the dark and subjective dark phase, and at the transition from the dark to the light phase during LD and at the presumptive transition in DD (Supplementary Fig. 4). However, the CI of neither LMA nor Tb was affected by the loss of glutamatergic signaling in the DMH.

As the DMH[Vglut2] ablation and the *Vglut2* mRNA deletion are chronic lesions that might be affected by compensatory mechanisms, we decided to test the effect of acute inhibition of the DMH[Vglut2] neurons on Cort, LMA and Tb rhythms. We injected the DMH in 9 *Vglut2-ires-Cre* mice with a viral vector containing a modified human alpha-1 glycine receptor (*AAV10-DIO-hGlyR-mCherry*) which is insensitive to glycine but has 100-fold increased sensitivity to the antiparasitic drug ivermectin (IVM) compared to the unmutated glycine receptor[16]. Because the half-life of IVM is about 72 hrs, the injection of IVM induces long-lasting neuronal inhibition (4–5 d). To allow the drug to achieve a stable concentration in the CNS, we examined Cort levels between 24 and 48 hrs after drug administration[17] compared with injection of vehicle. We administered vehicle (VEH) or IVM (5 mg/kg, ip) one hour after the light or subjective light onset and sampled the mice the next day at ZT1 and ZT13 to evaluate the Cort levels; one week later, they received the opposite treatment. In 5 mice in which expression of mCherry was seen in at least 70% of the DMH bilaterally (Fig. 2k, l, Supplementary Fig. 5a), the Cort levels at ZT13 under LD were reduced from 31.7 ± 2.3 ng/ml after VEH to 15.6 ± 4.8 ng/ml after IVM ($p = 0.006$), and under DD from 28.1 ± 6.9 ng/ml after VEH to 9.3 ± 1.8 ng/ml after IVM ($p = 0.001$, see Fig. 2m). The inhibition of the DMH[Vglut2] neurons with IVM reduced the Cort CI by 58.9 ± 15.9% in LD (p = 0.01) and by 92.3 ± 9.9% in DD ($p = 0.019$, see Fig. 2n). The average CI of LMA after IVM administration was reduced by 103.3 ± 21.1% in LD (i.e., the day-night difference was reversed due to higher levels of LMA during the light period) ($p = 0.011$, Supplementary Fig. 5b–d), and was reduced in DD by 81.3 ± 9.1% when compared with vehicle ($p = 0.016$, Supplementary Fig. 5e–g) mainly due to the shifting of much of the LMA peak from the early dark period to the light period, however, the total LMA was not significantly different at 24–48 h after IVM both in DD and LD (Supplementary Fig. 5b–g). IVM also induced a decrease in Tb during the dark period, with a reduction of 0.5 ± 0.2 °C observed in

LD compared with vehicle administration ($p = 0.024$, Supplementary Fig. 5h–j). In DD conditions, the decrease in Tb during presumptive night was 0.4 ± 0.1 °C ($p = 0.014$, Supplementary Fig. 5k–m). Tb and LMA returned to normal around day 5 after IVM either in LD or DD. The Tb CI was reduced in LD by 38.4 ± 6.2% after administration of IVM ($p = 0.025$, Supplementary Fig. 5j). In all the 3 experiments (DTA, iCre and hGlyR), the anatomical controls (animals with injections that missed the target) failed to show the responses observed in LMA or Tb.

These data clearly show that glutamatergic transmission by the DMH[Vglut2] neurons plays an important role in the increase of Cort levels at the beginning of the active phase. The ablation or inhibition of DMH[Vglut2] neurons reduces the peak of Cort to the early morning levels. The DMH[Vglut2] neurons also increase LMA in a crepuscular pattern (at the transitions between the light and dark periods) and their loss causes a small (approximately 0.3–0.5 °C) decrease in Tb, but has little if any effect on the circadian rhythm of Tb.

## Glutamatergic DMH neurons send monosynaptic projections to the PVH[CRH] neurons and boost Cort levels

As the ablation of DMH[Vglut2] neurons reduced the circadian peak of Cort levels, we hypothesized that activation of these neurons may elevate the Cort levels in preparation for the active phase. First, we injected the DMH of 15 *Vglut2-ires-Cre* mice with *AAV10-EF1α-DIO-hM3Dq-mCherry* (Fig. 3a, b). Five to six weeks later, the mice were injected i.p. with clozapine-N-oxide (CNO) 0.3 mg/kg at ZT3. The acute activation of DMH[Vglut2] neurons caused an increase in Cort levels from 7.4 ± 3.7 ng/ml just before the animals received the dose of CNO, to 162.5 ± 16.4 ng/ml 1 h after the administration of CNO ($p < 0.001$, see Fig. 3c). This increase is around three times higher than the Cort levels detected in the same mice that received saline or in WT mice injected with CNO. These results confirm that activation of the DMH[Vglut2] neurons boosts Cort release.

To determine whether the DMH[Vglut2] neurons make direct synaptic contacts onto the PVH[CRH] neurons to activate them and so to increase Cort levels, we conducted conditional anterograde tracing and conditional monosynaptic retrograde rabies tracing studies. We first evaluated the projection pattern of DMH[Vglut2] neurons using *Vglut2-ires-Cre* mice crossed with reporter mice that express the fluorescent protein Venus (green) in the CRH expressing neurons (*Vglut2-ires-Cre::CRH-Venus*). Following injection of *AAV8-DIO-ChR2-mCherry* into the DMH, we found a dense projection from DMH[Vglut2] neurons to the PVH that formed appositions with the PVH[CRH] neurons (Fig. 3d, e). To identify the locations of DMH[Vglut2] neurons potentially forming direct connections with the PVH[CRH] cells, we injected the PVH of 5 *CRH-ires-Cre* mice with an AAV expressing TVA avian receptor (*AAV8-Ef1a-DIO-TVA-mCherry*) and an AAV expressing rabies glycoprotein necessary for rabies viral transfection (*AAV8-CAG-DIO-rabiesG*) specifically in the PVH[CRH] neurons. Twenty-one days later, to label and map the direct monosynaptic inputs to the PVH[CRH] neurons, we injected a glycoprotein (G)-deleted rabies virus (RVdG) that expresses EGFP and is enveloped with the avian ASLV type A protein (EnvA), which utilizes the TVA receptor for cell entry (*EnvA-ΔG-rabies-EGFP*). This results in PVH[CRH] neurons infected by both AAV and rabies viruses (starter cells) displaying both green and red fluorescent signals, while neurons expressing only EGFP-Rabies (green) were retrogradely labeled by the viral particles produced in the starter cells (Supplementary Fig. 6a, b). Transneuronally infected neurons were seen in areas previously reported to project to the PVH such as the bed nucleus of the stria terminalis, preoptic area, arcuate nucleus, and DMH, and we also observed a large number of retrogradely labeled neurons within the PVH itself and its immediate surrounding areas (Supplementary Fig. 6c, d)[18]. We found few, if any, EGFP-Rabies infected neurons in the SCN, which supports the hypothesis that circadian control of Cort secretion by the SCN requires intermediate relays such as the SPZ and DMH.

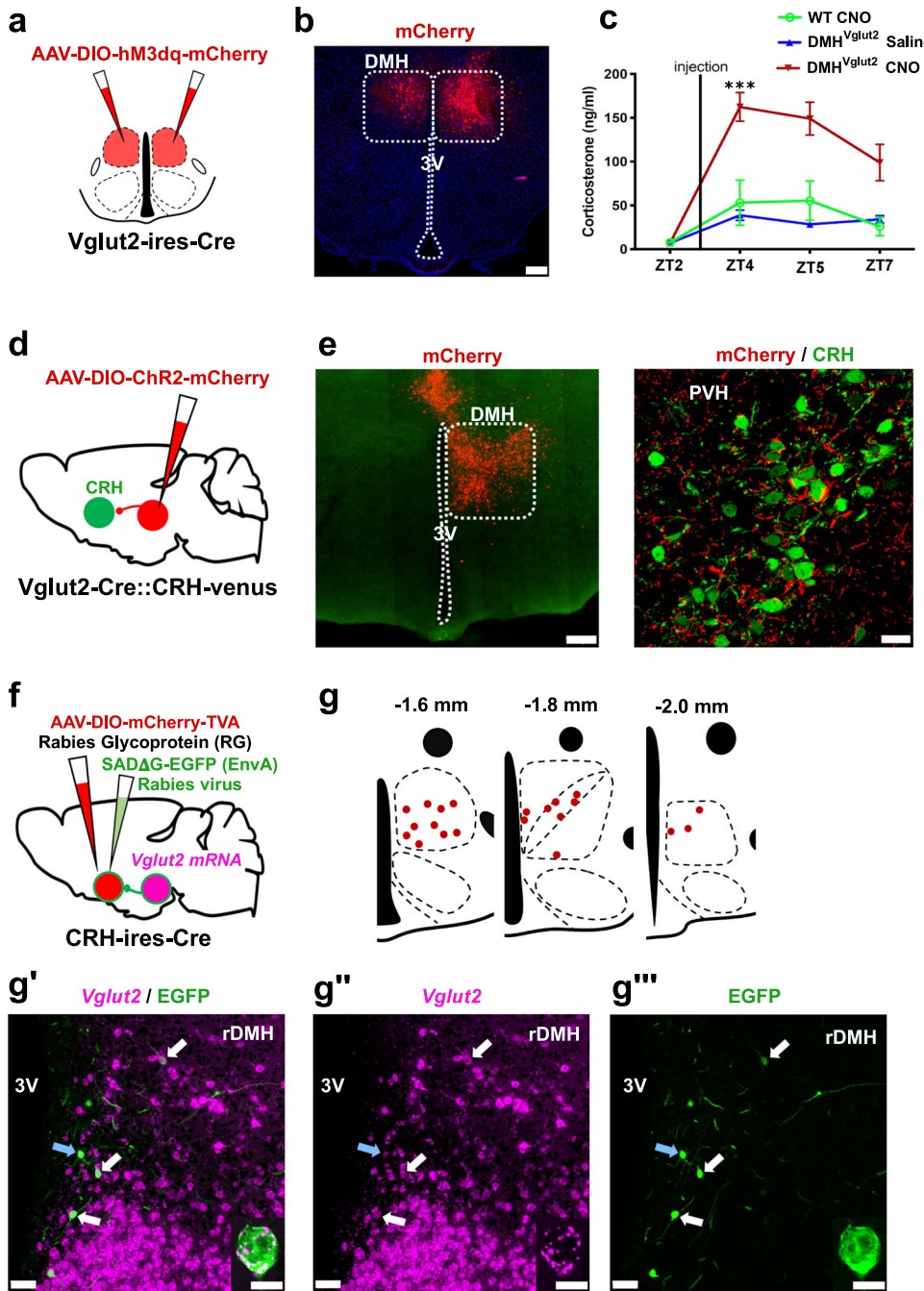

**Fig. 3 | DMH$^{Vglut2}$ neurons project to the CRH neurons of the PVH and their chemogenetic activation increases Cort. a** Schematic of *AAV-DIO-hM3dq-mCherry* injection targeting the DMH$^{Vglut2}$ neurons. **b** Representative image of an *AAV-DIO-hM3dq-mCherry* (in red) injection site in the DMH. Representative of 15 cases. **c** CNO-mediated chemogenetic-stimulation of DMH$^{Vglut2}$ neurons boosted Cort to levels much higher than the usual daily peak (cf. Fig. 2c; *Two-way ANOVA; Tukey's* multiple comparisons test. CT4 WT CNO [$n = 7$] vs DMH$^{Vglut2}$ CNO [$n = 15$]: **$p = 0.0016$, CT4 DMH$^{Vglut2}$ Saline vs DMH$^{Vglut2}$ CNO: ***$p < 0.0001$; $n = 15$, mean and ± SEM). **d** Schematic of AAV-DIO-ChR2-mCherry injection targeting DMH$^{Vglut2}$ neurons. **e** Representative photomicrograph of the DMH$^{Vglut2}$ neurons expressing ChR2-mCherry (left panel, in red), and their appositions with PVH$^{CRH}$ neurons (right panel, in green). Representative of 3 cases. **f** Schematic of the EnvA-rabies

experiment to map the monosynaptic input from the DMH$^{Vglut2}$ neurons to PVH$^{CRH}$ neurons. **g** Mapping of the rabies-Vglut2 co-labeling distribution at different rostro-caudal levels of the DMH, and representative images showing Vglut2 mRNA expression (**g'** and **g''**, in magenta) and rabies expression (**g'** and **g'''**, in green) within the rostral DMH (rDMH). Representative of 2 independent experiments. The arrows point to the co-labeled cells, and the inset in the lower-right part shows a higher magnification of the neuron pointed by the blue arrow. The hM3dq and ChR2 signal were enhanced with immunofluorescence for mCherry, while the Rabies infected cells were enhanced using an EGFP antibody. Reference scale bar: in (**b**, **e**) (left)= 200 μm, (**e**) (center) and (**g'–g'''**) = 50 μm, in (**g'–g''**) insets = 10 μm. 3 V, third ventricle. Atlas levels correspond to Paxinos and Franklin Atlas[52].

We also combined rabies tracing from PVH$^{CRH}$ neurons with in situ hybridization for *Vglut2* mRNA. We found about half of retrogradely-EGFP labeled neurons in the DMH expressed *Vglut2*, confirming that DMH$^{Vglut2}$ neurons are likely to directly innervate PVH$^{CRH}$ cells. In particular, we observe that the largest number of retrogradely labeled neurons expressing *Vglut2* were located in the anterior portion of the DMH (Fig. 3f, g). By contrast, retrogradely labeled neurons that expressed *Vgat* mRNA were found throughout the DMH, but with a

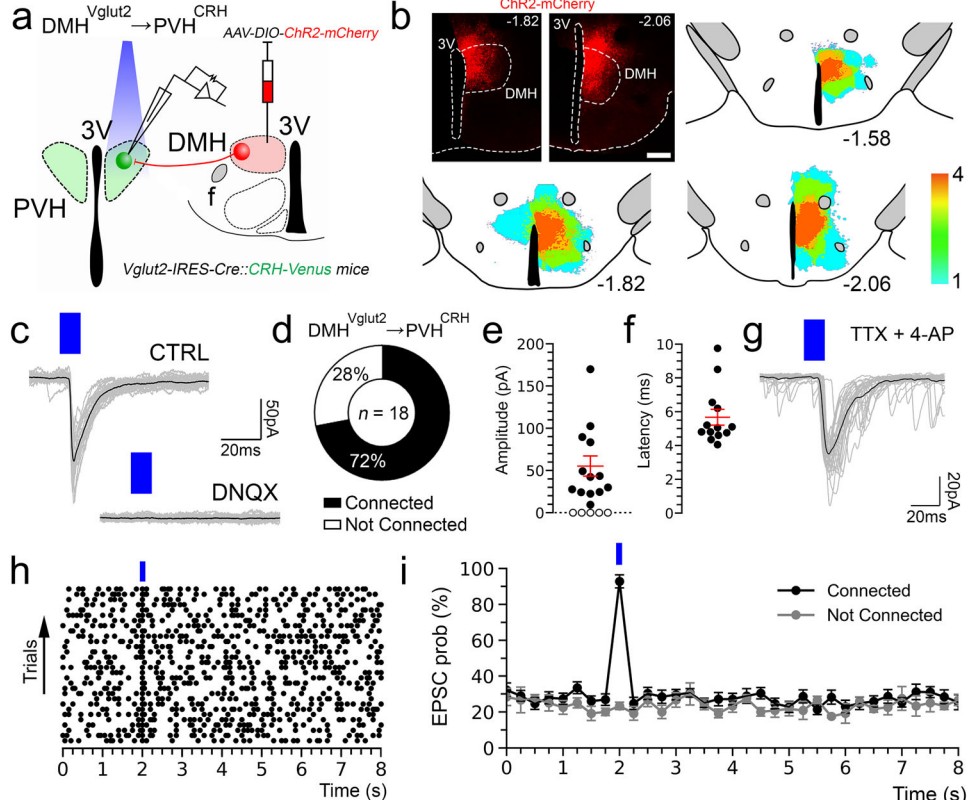

**Fig. 4 | In vitro optogenetic stimulation of the DMH glutamatergic input directly excites PVH$^{CRH}$ neurons. a** Schematic representation of the recording to test the proposed DMH$^{Vglut2}$ → PVH$^{CRH}$ circuit; *Vglut2-ires-Cre::CRH-Venus mice* were injected with *AAV-DIO-ChR2-mCherry* in the DMH and recordings were conducted in brain slices from Venus-labeled PVH$^{CRH}$ neurons while photostimulating the ipsilateral DMH$^{Vglut2}$ input (the DMH is shown on the opposite side of the brain here for ease of illustration). **b** An example of ChR2-mCherry native expression after an injection in the DMH (*top left*) and density plots of the *AAV-DIO-ChR2-mCherry* injection sites (*n* = 4 mice; *right* and *bottom*). **c** AMPA-mediated opto-evoked excitatory post-synaptic currents (oEPSCs) recorded in PVH$^{CRH}$ neurons (*upper trace*) and blockade by AMPA receptor antagonist DNQX, 200 μM (*lower trace; n* = 4, from 2 mice). **d** Percentages of PVH$^{CRH}$ neurons responding (*Connected*) and not responding (*Not Connected*) to photostimulation of the DMH$^{Vglut2}$ input (*n* = 18

PVH$^{CRH}$ recorded neurons, from 4 mice). **e** Amplitude (filled markers, cells responding to photostimulation, *n* = 13, open markers, cells not responding to photostimulation, *n* = 5 neurons from 4 mice; mean and ± SEM of responding neurons) and (**f**) latency of oEPSCs in PVH$^{CRH}$ neurons in response to photostimulation of the DMH$^{Vglut2}$ input (mean and ± SEM; *n* = 13 neurons from 4 mice). **g** oEPSCs in PVH$^{CRH}$ neurons recorded in TTX 1 μM + 4-AP 500 μM (*n* = 6 neurons from 2 mice) indicating monosynaptic connectivity. **h** Raster plot of EPSCs in a representative PVH$^{CRH}$ neuron with photostimulation of the DMH$^{Vglut2}$ → PVH$^{CRH}$ input (bin duration: 50 ms). **i** EPSC probability in response to photostimulation of the DMH$^{Vglut2}$ → PVH$^{CRH}$ input (black, *n* = 13 neurons; grey, *n* = 5 neurons, from 4 mice, mean and ± SEM). Reference scale bar: in b = 250 μm. f, fornix, 3 V, third ventricle. Atlas levels correspond to Paxinos and Franklin Atlas[52].

more caudal predominance (Supplementary Fig. 6e, f), indicating that there is also a direct GABAergic input predominantly from the caudal DMH to the PVH$^{CRH}$ neurons (see below).

## Stimulation of the DMH$^{Vglut2}$ neurons directly excites PVH$^{CRH}$ neurons

To test the input from the DMH$^{Vglut2}$ neurons to the PVH$^{CRH}$ neurons (DMH$^{Vglut2}$ → PVH$^{CRH}$) functionally, we conducted in vitro channelrhodopsin-2(ChR2)-assisted circuit mapping (CRACM) recordings. We injected the DMH with an *AAV8-DIO-ChR2-mCherry* in a new cohort of 4 *Vglut2-ires-Cre::CRH-Venus* mice, and then three to four weeks later we recorded from Venus-labeled PVH$^{CRH}$ neurons in brain slices while photo-stimulating ipsilateral axons and terminals of DMH$^{Vglut2}$ neurons within the PVH (Fig. 4a, b). All recordings were performed in the PVH ipsilateral to the injection site during the daytime (ZT3-8), at the nadir of the daily Cort cycle[19]. Optogenetic stimulation of the DMH$^{Vglut2}$ input evoked excitatory synaptic responses in 72% of the PVH$^{CRH}$ neurons recorded (*n* = 18). This effect was mediated by the release of glutamate and AMPA receptor signaling as the opto-evoked excitatory post-synaptic currents (oEPSCs) were blocked by the AMPA receptor antagonist DNQX (*n* = 4; Fig. 4c–f). Furthermore, the probability of oEPSCs in the PVH$^{CRH}$ cells peaked about 5 msec after the

photostimulation of the DMH$^{Vglut2}$ terminals and the oEPSCs persisted in the presence of TTX (TTX 1 μM) and 4-aminopyridine (4-AP, 200–500 μM) indicating monosynaptic connectivity (*n* = 6 out of 8; Fig. 4f, g). These experiments demonstrate that DMH$^{Vglu2}$ neurons have robust direct synaptic input to PVH$^{CRH}$ neurons (Fig. 4h, i) that is consistent with DMH$^{Vglut2}$ neurons driving the elevation of Cort levels during the early active period (Fig. 2).

## Ablation, inhibition, or disruption of GABAergic signaling in DMH$^{Vgat}$ neurons reduces the daily peak in Cort secretion only under constant darkness

Although inputs from DMH$^{Vglut2}$ neurons to PVH$^{CRH}$ cells are necessary to drive the daily Cort increase, their ablation does not completely eliminate the circadian rhythm of Cort secretion, as is seen with non-specific DMH ablation[12]. It has been estimated that half of the synapses to the PVH$^{CRH}$ neurons are GABAergic[20], and our rabies virus experiment showed that about half of the DMH cells innervating the PVH$^{CRH}$ neurons are GABAergic. Thus, we decided to evaluate whether the DMH$^{Vgat}$ neurons could play a complementary role in the circadian release of Cort. We hypothesized that the DMH$^{Vgat}$ neurons could contribute to the circadian rhythms of Cort secretion either by inhibiting PVH$^{CRH}$ neurons during the inactive cycle to suppress Cort

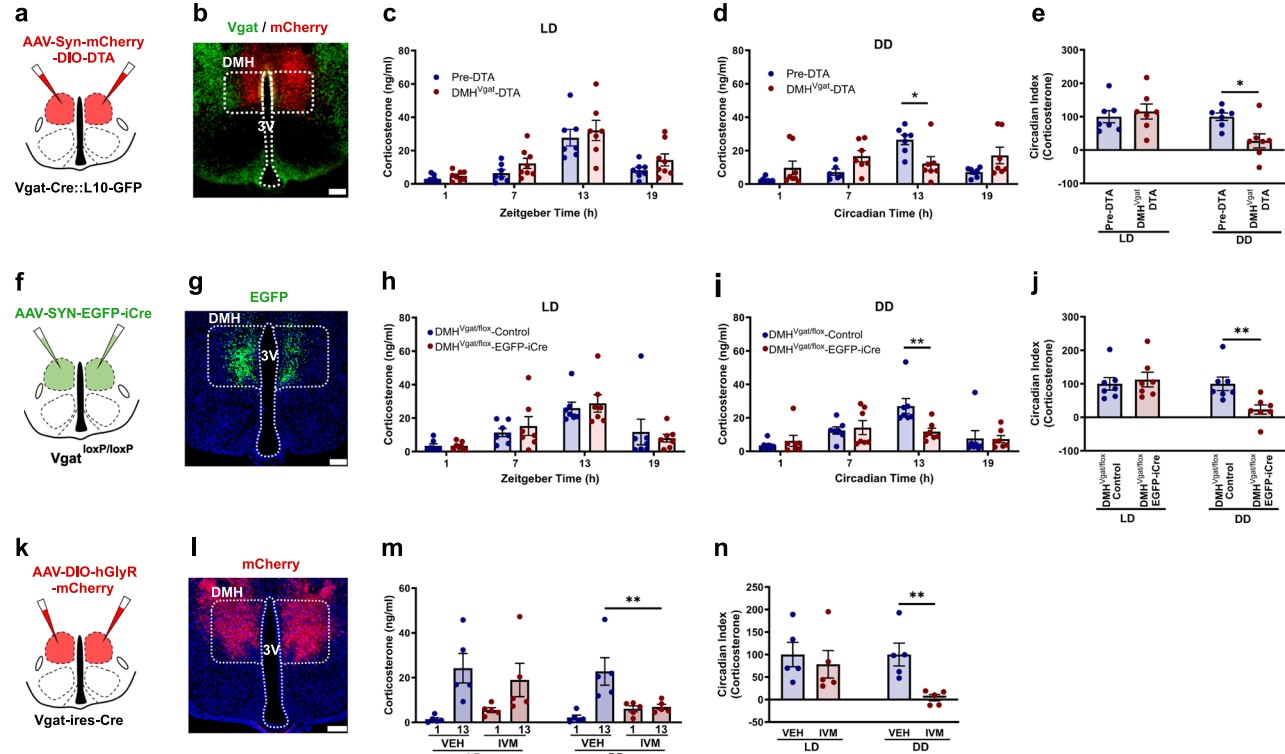

**Fig. 5 | Ablation of DMH^Vgat neurons reduces the endogenous circadian peak of Cort release only in constant darkness. a** Schematic representation of DMH^Vgat neuron ablation. **b** Representative micrograph showing GABAergic neurons (in green) and *AAV-DIO-DTA-mCherry* (in red) in the DMH of a *Vgat-ires-Cre::L10-GFP* mice. There were no remaining GABAergic neurons in the region of mCherry expression. Representative of 8 cases. **c** DMH^Vgat neuron ablation did not alter the daily peak in Cort release in LD, (**d**) but eliminated it almost entirely under DD (*Two-way ANOVA*; *Tukey's* multiple comparisons test. CT13 Pre-DTA vs DMH^Vgat-DTA: *\*p = 0.014, n = 8*. **e** The Cort CI was not significantly changed by DMH^Vgat ablation in LD, but was reduced by 72.4 ± 21.3% in DD. (*Paired t-test, Two-tailed*: t = 3.028, df = 12, *\*p = 0.01, n = 8*). **f** Schematic representation of *Vgat* gene deletion in the DMH. **g** Representative photomicrograph showing the EGFP expression in *Vgat*-deleted neurons (green). Representative of 7 cases. **h** *Vgat* deletion from DMH neurons did not change the circadian regulation of Cort release in LD, (**i**) but eliminated the daily peak of Cort at CT13 in DD (*Two-way ANOVA*; *Tukey's* multiple

comparisons test. CT13 DMH^Vgat/flox-Control vs DMH^Vgat/flox-EGFP-iCre: *\*\*p = 0.006, n = 7*). **j** The CI of Cort secretion was not affected by *Vgat* gene deletion in the DMH in LD but was reduced by 76.6 ± 13.6% in DD in DMH^Vgat-deleted mice (*Unpaired t-test, Two-tailed*: t = 3.160, df = 12, *\*\*p = 0.008, n = 7*). **k** Schematic representation of the *AAV-DIO-hGlyR-mCherry* injection in the DMH of *Vgat-ires-Cre* mice. **l** Representative micrograph of *hGlyR-mCherry* expression (in red) in the Vgat neurons of the DMH. Representative of 5 cases. **m** IVM chemo-inhibition of the DMH^Vgat neurons by hGlyR did not alter the Cort levels in LD, but prevented the peak at CT13 in DD (*Two-way ANOVA*; *Tukey's* multiple comparisons test. CT13 VEH vs IVM: *\*\*p = 0.006, n = 5*). **n** The Cort CI was reduced by 95.7 ± 6.9% in DD (*Paired t-test, Two-tailed*: t = 3.641, df = 8, *\*\*p = 0.006, n = 5*). In all cases we visualize the native signal except for the hGlyR signal that was enhanced with immunofluorescence for mCherry. Reference scale bar = 200 μm; 3 V, third ventricle. Data are presented as mean ± SEM.

secretion, or by dis-inhibiting PVH^CRH neurons (i.e., inhibiting inhibitory inputs to them) during the active cycle. Thus, we used *Vgat-ires-Cre* mice crossed with a Cre-dependent GFP reporter mouse line *R26-loxSTOPlox-L10-GFP* to generate *Vgat-ires-Cre::R26-loxSTOPlox-L10-GFP* mice (*Vgat-ires-Cre::L10*), in which Vgat neurons expressing Cre recombinase show green fluorescence. We collect blood samples at 4 temporal points in LD (ZT 1, 7, 13, 19), and then in the same temporal points in DD (CT 1, 7, 13, 19), as we did with the *Vglut2-ires-Cre* mice. Also, we recorded LMA and Tb for 12 days in LD, and then 12 days in DD. Then, we injected the DMH of *Vgat-ires-Cre::L10* mice with an *AAV10-hSyn-mCherry-DIO-DTA* to ablate the DMH^Vgat neurons. Four weeks after the injections, we recorded and took blood samples from the mice in LD and DD conditions. From a total of 13 *Vgat-ires-Cre::L10* mice injected, we analyzed data from 8 mice in which a blinded investigator found that the injection sites covered at least 70% of the DMH bilaterally (Fig. 5a, b, Supplementary Fig. 7a, b). DMH^Vgat ablation did not alter the daily rhythm of Cort release in LD but reduced the CT13 peak under DD from 26.5 ± 2.9 ng/ml in Ctrl to 14.1 ± 4.4 ng/ml (*p = 0.014*, Fig. 5c, d). Similarly, there was no statistically significant change in the CI of Cort in LD, but it was reduced by 72.4 ± 21.3% in DD after ablation of DMH^Vgat neurons (*p = 0.01*, Fig. 5e). Importantly, ablation of DMH^Vgat neurons did not change the levels of Cort at its nadir (ZT1 or CT1),

which might have been expected if DMH^Vgat neurons with direct inputs to PVH^CRH cells contributed to circadian rhythms of Cort by suppressing secretion during the daily nadir. To test the effect of ablation of DMH^Vgat neurons on other aspects of Cort secretion, we also measured Cort levels after one hour of restraint stress, but these were unaffected by the ablations (Supplementary Fig. 7c). In contrast to the effects on Cort rhythms, LMA was dramatically reduced during the dark phase of LD after DMH^Vgat ablation, the total count over the dark period decreasing from 873.5 ± 45 counts to 429.5 ± 52.5 counts (*p < 0.001*); LMA was reduced even further in DD during the presumptive dark phase from 771.2 ± 40.6 counts to 401.7 ± 44.9 counts (*p < 0.001*, Supplementary Fig. 7h), resulting in a reduction in the Circadian Index of LMA by 38.23 ± 9.2% in LD (*p = 0.005*) and by 50.61 ± 12.9% in DD (*p = 0.003*, Supplementary Fig. 7i, j), and a significant reduction in the amplitude of the peak at 24 h in the periodogram and in the cosinor analysis of LMA in both LD and DD (LD: *p < 0.001*; DD: *p < 0.001*, Supplementary Fig. 7e, g). Thus, most of the reduction in LMA after DMH lesions appears to be due to loss of DMH^Vgat neurons. In the mice with ablation of DMH^Vgat neurons, Tb was about 0.3 °C lower throughout the day in LD and during the subjective dark period in DD, but there was no statistically significant change in Tb in either the CI or cosinor analysis (Supplementary Fig. 7l–s).

We interpreted these results as indicating that it was unlikely that DMH[Vgat] neurons played a role in suppressing Cort secretion during the inactive (light) phase, but that they potentially accounted for the other half (not caused by the DMH[Vglut2] neurons) of the surge in Cort levels during the active (dark) phase. We next evaluated whether this effect was due to GABA release from these neurons (vs. some other neurotransmitter in the same neurons). Thus, we placed *AAV8-SYN-EGFP-iCre* or *AAV8-DIO-EGFP* injections in the DMH in *Vgat[loxP/loxP]* (*Vgat-flox*) mice to delete expression of functional Vgat protein and so to abolish GABA transmission from the GABAergic neurons of the DMH. From 12 mice injected with AAV-EGFP-iCre in the DMH, 7 with bilateral injections covering at least 70% of the DMH were included in the analysis (Fig. 5f, g, Supplementary Fig. 8a). *Vgat* deletion from DMH neurons did not change the circadian regulation of Cort release in LD, but again reduced the CI of Cort release in DD by $76.6 \pm 13.6\%$ ($p = 0.008$, Fig. 5j), by reducing the CT13 peak of Cort levels in DD from $27.0 \pm 4.5$ ng/ml in controls to $11.7 \pm 2.1$ ng/ml in *Vgat*-deleted mice ($p = 0.006$, Fig. 5h). Thus, most if not all of the effect of the DMH[Vgat] neurons on the circadian release of Cort is mediated by GABA. Interestingly, although the total levels of LMA during the dark period were decreased by Vgat deletion in the DMH (from $979.2 \pm 66.8$ counts in the control group to $681.5 \pm 55.6$ counts in the deletion group during LD; $p < 0.001$, Supplementary Fig. 8f), the reduction was not nearly as profound as seen after the ablation of the DMH[Vgat] neurons, suggesting that other transmitters released from those neurons could play a role in regulating LMA. Results in DD were similar (from $979.2 \pm 66.8$ total counts in DMH[Vgat/flox]-Control group to $681.5 \pm 55.6$ counts in the DMH[Vgat/flox]-EGFP-iCre during DD; $p < 0.001$, Supplementary Fig. 8f), so that the CI of LMA was reduced by only $25.7 \pm 6\%$ in LD ($p = 0.001$) and $33.3 \pm 12.6\%$ in DD ($p = 0.021$, Supplementary Fig. 8g), similar to the effect observed in the amplitude in the cosinor analysis (Supplementary Fig. 8g, h). Tb of DMH[Vgat]–deleted mice was also reduced during the dark phase by $0.2\,°C$ in LD (mostly between ZT14-18; $p = 0.001$, Supplementary Fig. 8j–n), and in the subjective dark by $0.3\,°C$ in DD ($p < 0.001$, Supplementary Fig. 8l–n). As a result, the Circadian Index of Tb was reduced in the DMH[Vgat]–deleted mice by $20.8 \pm 8.8\%$ in LD ($p = 0.037$) and by $26.6 \pm 8.9\%$ in DD ($p = 0.012$, Supplementary Fig. 8o), similar to the reduction in the amplitude in the cosinor analysis (Supplementary Fig. 8o, p).

To determine whether these chronic conditional lesions or impairments of GABA transmission might be affected by compensatory mechanisms, we examined the effect of acute inhibition of the DMH[Vgat] neurons on the circadian release of Cort. We placed injections of *AAV-DIO-hGlyR-mCherry* in the DMH of *Vgat-ires-Cre* mice. From 7 mice injected, 5 mice with bilateral injection covering at least 70% of the DMH were included for analysis (Fig. 5k, l, Supplementary Fig. 9a). After 4 weeks allowing the full expression of hGlyR, mice were injected with either vehicle or IVM (5 mg/Kg, ip) at ZT1 in LD, or CT1 in DD, and Cort levels were sampled 24 h and 36 h later. The inhibition of the DMH[Vgat] neurons did not change the Cort levels under LD, but reduced the level at CT13 from $22.8 \pm 6.1$ ng/ml after vehicle to $6.9 \pm 1.1$ ng/ml after IVM administration ($p = 0.006$, Fig. 5m) and reduced the CI by $95.7 \pm 6.9\%$ in DD ($p = 0.006$, Fig. 5n). Similar to the effects of their ablation, the inhibition of DMH[Vgat] neurons reduced LMA during the dark or subjective dark period between 24–48 h after IVM administration, reducing the CI in LD by $77.7 \pm 5.7\%$ (VEH $p = 0.001$, Supplementary Fig. 9b–d), and in DD by $80.7 \pm 5\%$ ($p = 0.002$, Supplementary Fig. 9e–g). The Tb during the dark period or presumptive dark period was lower after IVM administration when compared vs VEH (which includes the effect of the handling in contrast to the baseline), therefore, the CI of Tb was significantly reduced in both LD and DD, 24–48 h after IVM administration by $77.7 \pm 8.1\%$ in LD ($p = 0.005$, Supplementary Fig. 9h–j), and by $68.3 \pm 14\%$ in DD ($p = 0.004$, Supplementary Fig. 9k–m). In all three experiments (ablation, deletion, and inhibition), the changes observed in levels or rhythms of LMA or Tb were not

observed in the anatomical controls (animals with injections that missed the DMH). These data suggest that DMH[Vgat] neurons participate in the circadian peak of Cort release at the beginning of the active period (as seen in DD), likely reducing the inhibitory tone to the PVH[CRH] neurons at this temporal point, but that in LD light can act as a cue for elevation of Cort at the onset of the active period, presumably through another pathway. The ablation of DMH[Vgat] neurons did not, however, increase the levels of Cort secretion at ZT1 or CT1, indicating that they do not shape circadian secretion of Cort by suppressing levels at their nadir. Also, the DMH[Vgat] neurons play a larger role than DMH[Vglut2] neurons in the circadian regulation of LMA, as the ablation of DMH[Vgat] but not the DMH[Vglut2] neurons dramatically reduced LMA during the dark or subjective dark periods.

## The activation of DMH GABAergic neurons increases Cort levels, whereas cvPVH GABAergic neurons constrain Cort levels

As the ablation of the DMH[Vgat] neurons prevented the peak in DD conditions, we tested whether the activation of those neurons can induce Cort release. We placed injections of *AAV8-DIO-hM3Dq-mCherry* in the DMH of *Vgat-ires-Cre* mice. From 8 mice injected, 6 mice were confirmed to have injection sites covering at least 70% of the DMH bilaterally (Fig. 6a, b). CNO administration at the beginning of the light phase (ZT2) elevated the Cort levels to $36.8 \pm 9$ ng/ml one hour after administration, while Cort was only $10.2 \pm 2$ ng/ml after vehicle ($p = 0.016$, Fig. 6c). Although activation of DMH[Vglut2] neurons produced a larger increase in Cort levels ( $>150$ ng/ml) than activation of DMH[Vgat] neurons, in both cases the increase in Cort levels caused by CNO was at least as great as the circadian peak of Cort in undisturbed littermates. Thus, activation of either DMH[Vgat] or DMH[Vglut2] neurons is capable of elevating Cort levels within the range achieved by circadian rhythms.

Because activation of the DMH[Vgat] neurons is expected to release GABA and inhibit their postsynaptic targets, we hypothesized that the DMH[Vgat] neurons induce Cort release by disinhibiting the PVH[CRH] neurons through inhibitory relay neurons, similar to the mechanism by which GABAergic inputs from the arcuate nucleus increase Cort secretion during food deprivation[18]. To understand the circuit through which DMH GABAergic neurons promote CRH secretion, we crossed *Vgat-ires-Cre* mice with *CRH-Venus* reporter mice to generate *Vgat-ires-Cre::CRH-Venus* mice. We then injected the DMH of 4 *Vgat-ires-Cre::CRH-Venus* mice with *AAV8-DIO-ChR2-mCherry*. These injections were relatively small (3–9 nl compared to 24–45 nl for the other experiments) and covering mostly the rostral part of the DMH (rDMH). We found that axons and terminals labeled anterogradely (expressing mCherry, in red) from rDMH[Vgat] neurons largely avoided PVH[CRH] neurons (green). The bulk of the labeled terminal field was in areas surrounding the PVH, as well as in the region medial and ventral to the PVH[CRH] neurons and extending into the caudal ventral PVH (Fig. 6d, e, j, k). Although the PVH contains very few GABAergic cells, many are found in the region surrounding the PVH, known as the peri-PVH area, as well as in the adjacent caudal ventral part of the PVH. As terminals from the DMH[Vgat] neurons blanket the caudal ventral PVH as well as the adjacent peri-PVH just outside it, for the sake of simplicity we will refer to this region as the cvPVH. Local inhibitory inputs within the PVH have been described from this region[21–23], so we decided to explore whether the cvPVH[Vgat] neurons might mediate disinhibition of the PVH[CRH] neurons during the circadian peak of Cort. To explore whether the cvPVH[Vgat] neurons might make monosynaptic contacts with PVH[CRH] neurons, we combined conditional retrograde rabies tracing from PVH[CRH] neurons with in situ hybridization for Vgat mRNA using the CRH-ires-Cre mice ($n = 2$). We observed doubly labeled neurons in the cvPVH suggesting monosynaptic GABAergic inputs from this region to the PVH[CRH] neurons (Fig. 6f, g). We next injected 3 *Vgat-ires-Cre::CRH-Venus* mice with *AAV8-DIO-ChR2-mCherry* in this part of the PVH. The injections covered the ventral and lateral region of the PVH including the cvPVH, and labeled axons ramified inside the PVH, including ones

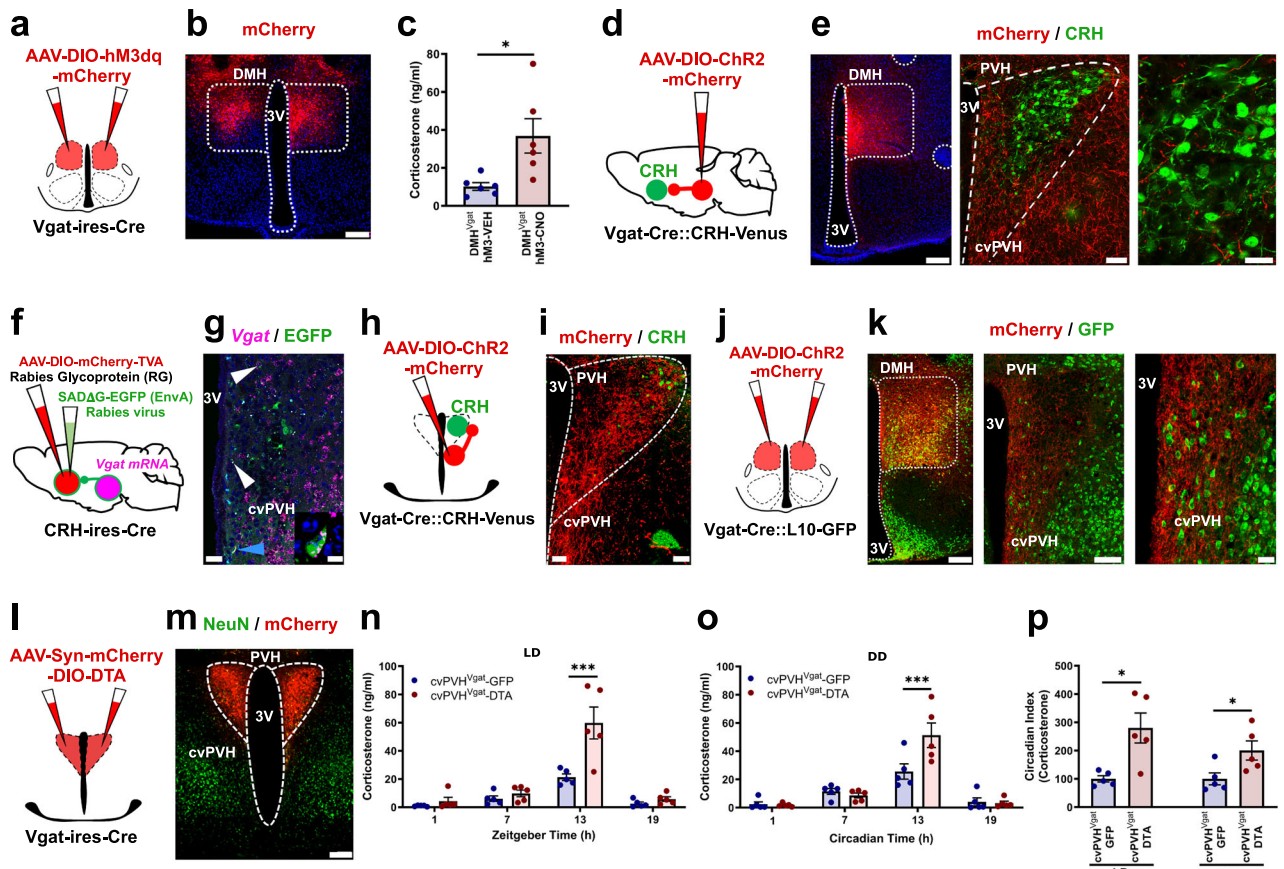

**Fig. 6 | Chemo-activation of DMH^Vgat neurons elevates Cort levels through cvPVH GABAergic neurons. a** Schematic of *AAV-DIO-hM3dq-mCherry* injections in the DMH in *Vgat-ires-Cre* mice. **b** Representative micrograph of hM3dq-mCherry expression (red) in the DMH. *n* = 6 independent experiments. **c** Chemogenetic activation of the DMH^Vgat neurons increased Cort levels from 10.2 ± 2 ng/ml after VEH to 36.8 ± 9 ng/ml after CNO administration (*Paired t-test, Two-tailed:* t = 2.875, df = 10, *\*p* = 0.016, *n* = 6). **d** Schematic of an *AAV-DIO-ChR2-mCherry* injection in the DMH of a *Vgat-Cre::CRH-Venus* mice. **e** Representative micrograph of rDMH^Vgat neurons expressing ChR2-mCherry (in red, *left* panel), and their axon pattern in the PVH (*central* panel). Note that rDMH^Vgat axon terminals mostly avoided the PVH^CRH neurons (in green, *right* panel) but were dense in the ventral-medial and caudal portion of the PVH (cvPVH). Representative of 4 cases. **f** Schematic of the EnvA-Rabies experiment to map monosynaptic inputs from the cvPVH^Vgat neurons to PVH^CRH neurons. **g** Representative photomicrograph showing the *Vgat mRNA* expression (in magenta) and the Rabies expression (in green) and doubly labeled neurons within the cvPVH. Arrows point to double labeled neurons; the blue arrow points to the neuron in the *lower-right* inset. Representative of 2 cases. **h** Schematic of an *AAV-DIO-ChR2-mCherry* injection in the cvPVH. **i** Representative image of the cvPVH^Vgat cell bodies expressing ChR2-mCherry (in red) in the ventral part of the PVH^CRH neuron field (in green). Their axons spread through the more dorsal PVH and form appositions with the PVH^CRH neurons (*lower-right* inset). Representative of

3 cases. **j** Schematic of the *AAV-DIO-ChR2-mCherry* injections in the DMH. **k** Micrograph showing the expression of ChR2-mCherry (in red) in the DMH^Vgat neurons (in green, *left* panel), and their projections and appositions to the cvPVH^Vgat neurons (*central and right* panel). Representative of 5 cases. **l** Schematic of the PVH^Vgat neuron ablation. **m** Representative image of the injection site showing mCherry expression (in red) in non-Vgat neurons in the PVH, and the loss of neurons detected by the reduction of NeuN (in green) in the cvPVH and along the PVH borders due to the ablation of the Vgat cells. *n* = 5 independent cases. **n** The cvPVH^Vgat ablation boosted the Cort increase at ZT13 in LD (*Two-way ANOVA; Tukey's* multiple comparisons test. ZT13 cvPVH^Vgat-EGFP vs cvPVH^Vgat-DTA: *\*\*\*p* < 0.0001, n = 5), and (**o**) at CT13 in DD (Two-way ANOVA; Tukey's multiple comparisons test. CT13 cvPVH^Vgat-EGFP vs cvPVH^Vgat-DTA: *\*\*\*p* = 0.0003, *n* = 5). **p** The Cort CI increased after the cvPVH^Vgat neuron ablation by 180.2 ± 52.9% in LD (Unpaired t-test, Two-tailed: *t* = 3.336, df = 8, *\*p* = 0.01, *n* = 5) and 100.4 ± 33.9% in DD (Unpaired t-test, Two-tailed: *t* = 2.512, df=8, *\*p* = 0.036, *n* = 5). The hM3dq and ChR2 signal, but not in the DTA experiments, were enhanced with immunofluorescence for mCherry, while the Rabies infected cells were enhanced using an EGFP antibody. Reference scale bar: in (**b, e, l, k, m** = 200 μm, (**e**) (center), (**g, k**) (center) = 50 μm, in (**e**) (right panel), **i** (right panel) and **k** (right panel) = 20 μm, in (**g, i**) (insets) = 5 μm; 3 V, third ventricle. Data are presented as mean ± SEM.

making appositions with the CRH-Venus neurons (Fig. 6h, i). We then investigated whether the DMH^Vgat neurons innervate the cvPVH^Vgat neurons. Following injections of *AAV8-DIO-ChR2-mCherry* into the rostral DMH of *Vgat-ires-Cre::L10-GFP* mice (*n* = 5) we found dense terminal labelling in apposition to the cvPVH^Vgat neurons, suggesting direct connectivity (Fig. 6j, k). These neuroanatomical findings support the hypothesis of a circuit where the DMH^Vgat neurons disinhibit PVH^CRH neurons through direct inhibition of the cvPVH^Vgat neurons (DMH^Vgat → cvPVH^Vgat → PVH^CRH).

To test this model, we ablated the Vgat expressing neurons in the cvPVH by placing injections of *AAV10-hSyn-mCherry-DIO-DTA* in the cvPVH of *Vgat-ires-Cre* mice (*n* = 5). In all cases, the injection sites

marked by mCherry expression involved the cvPVH, where Vgat cells projecting to the PVH^CRH neurons are found, and the body of the PVH, where few Vgat cells are seen. (Fig. 6l, m, Supplementary Fig. 10a). After four weeks to allow full expression of DTA, we recorded LMA and Tb in the mice for 12 days in both LD and DD, and we collected blood samples for Cort measurements toward the end of both the LD and DD periods. The cvPVH^Vgat ablated mice showed higher levels of Cort but only at the beginning of the active phase, from 21.3 ± 2.3 ng/ml in controls to 59.8 ± 11.3 ng/ml in cvPVH^Vgat ablated mice in LD (*p* < 0.001, Fig. 6n), and from 25.6 ± 5.4 ng/ml to 51.3 ± 8.7 ng/ml in DD (*p* < 0.001, Fig. 6o). The CI of Cort also increased by 180.2 ± 52.9% in LD (*p* = 0.01) and 100.4 ± 33.9% in DD (*p* = 0.036, Fig. 6p). At the end of the protocol,

we also exposed the mice to restraint stress and found that the Cort levels were significantly higher in the cvPVH$^{Vgat}$ ablated mice ($p = 0.047$, Supplementary Fig. 10b). No major changes were observed in LMA or Tb with ablation of cvPVH$^{Vgat}$ neurons (Supplementary Fig. 10c–r). These data indicate that the cvPVH$^{Vgat}$ neurons play an important role in inhibiting Cort secretion, even limiting its peak during the daily surge in Cort secretion at CT13 and in response to restraint stress. However, these neurons do not appear to play a role in suppressing Cort levels at their daily nadir. DMH$^{Vgat}$ inhibition of cvPVH$^{Vgat}$ neurons partially disinhibits the PVH$^{CRH}$ neurons, allowing the surge in Cort secretion at CT13, but completely eliminating the cvPVH$^{Vgat}$ neurons allows even greater stimulation of the PVH$^{CRH}$ cells at that time by DMH$^{Vglut2}$ inputs.

### Dissecting the synaptic inputs of the DMH$^{Vgat}$→ cvPVH$^{Vgat}$ → PVH$^{CRH}$ circuit

To dissect the synaptic mechanisms by which the DMH$^{Vgat}$ neurons control the PVH$^{CRH}$ neurons, we injected the DMH of a new cohort of seven *Vgat-ires-Cre::CRH-Venus* mice with *AAV8-DIO-ChR2-mCherry*. We then performed patch clamp recordings in brain slices from green-labelled PVH$^{CRH}$ neurons on the same side of the brain while photo-activating the red DMH$^{Vgat}$ terminals in the PVH (Fig. 7a). Our rabies virus tracing data indicated that PVH$^{CRH}$ neurons receive a direct GABAergic input from the DMH but mainly from its caudal portion (Supplementary Fig. 6e, f). In support of these results, we observed that in three mice where the injection site was restricted mostly to the rostral part of the DMH (rDMH), the photostimulation of the DMH$^{Vgat}$ input evoked inhibitory synaptic responses in only 4 out of 17 recorded PVH$^{CRH}$ neurons (23%, $n = 17$; Fig. 7a–e, r), whereas in four mice in which ChR2 expression involved the more caudal portion of the DMH, the photostimulation of the DMH$^{Vgat}$ input evoked oIPSCs in nearly all PVH$^{CRH}$ neurons (96%, $n = 23$; Supplementary Fig. 11a–g). As the direct GABAergic inputs from the caudal DMH to PVH$^{CRH}$ neurons do not appear to play a role in inhibiting the PVH$^{CRH}$ neurons at the circadian nadir of Cort, we focused on whether the rDMH GABAergic inputs could disinhibit the PVH$^{CRH}$ neurons at the daily peak of Cort secretion.

To investigate whether rDMH$^{Vgat}$ neurons directly inhibit the cvPVH$^{Vgat}$ neurons, we expressed *AAV8-DIO-ChR2-mCherry* in the rDMH of 5 *Vgat-ires-Cre::L10-GFP* mice. We then recorded from GFP-labelled cvPVH$^{Vgat}$ neurons while stimulating the ipsilateral putative rDMH$^{Vgat}$ input to them (rDMH$^{Vgat}$ →cvPVH$^{Vgat}$) (Fig. 7f). Photostimulation evoked opto-inhibitory postsynaptic currents (oIPSCs) in 14 out of 15 recorded cvPVH$^{Vgat}$ neurons. These oIPSCs depended on GABA$_A$ receptor-mediated signaling, as they were blocked by bicuculline ($n = 4$; Fig. 7g–m). Furthermore, oIPSCs persisted in the presence of TTX ($1\,\mu M$) and 4-AP ($200\,\mu M$) indicating monosynaptic connectivity ($n = 6$, Fig. 7n).

When we activated the rDMH$^{Vgat}$ terminals at different frequencies while recording from the PVH$^{CRH}$ neurons in which single light pulses did not evoke oIPCS, we found that photostimulation of the ipsilateral rDMH$^{Vgat}$ input at 5 or 10 Hz caused a reduction in the number of spontaneous IPSCs, which became statistically significant at 20 Hz ($n = 7$; Fig. 8a–e). These results indicate that the rDMH$^{Vgat}$ neurons can disinhibit most PVH$^{CRH}$ neurons by reducing their GABAergic afferent input.

Our conditional tracing study and CRACM recordings suggest that the rDMH$^{Vgat}$ input disinhibits the PVH$^{CRH}$ neurons at least in part through the GABAergic neurons located in the cvPVH. We therefore tested whether the cvPVH$^{Vgat}$ neurons can directly inhibit the PVH$^{CRH}$ neurons (cvPVH$^{Vgat}$ → PVH$^{CRH}$). We placed *AAV8-DIO-ChR2-mCherry* injections in the cvPVH of 3 *Vgat-ires-Cre::CRH-Venus mice*. We then recorded from labelled PVH$^{CRH}$ neurons while photostimulating cvPVH$^{Vgat}$ neurons and terminals. Photostimulation of cvPVH$^{Vgat}$ neurons evoked oIPSCs in all PVH$^{CRH}$ neurons recorded ($n = 14$). These oIPSCs were mediated by GABA$_A$ receptor signaling as they were

blocked by bicuculline ($n = 4$) and were resistant to TTX + 4-AP, indicating monosynaptic connectivity ($n = 6$) (Fig. 8f–l). Interestingly, in four cases where the ChR2 expression was more dorsal and lateral involving the portion of the pPVH just lateral to the PVH only about half of the PVH$^{CRH}$ neurons we recorded from showed oIPSCs ($n = 4$) (Fig. 8m–r).

Taken together, these results indicate that rostral DMH$^{Vgat}$ (rDMH$^{Vgat}$) neurons can disinhibit PVH$^{CRH}$ neurons through GABAergic interneurons located in the cvPVH (rDMH$^{Vgat}$ → cvPVH$^{Vgat}$ → PVH$^{CRH}$), promoting the circadian increase of Cort levels at CT13.

## Discussion

This study identifies a circuit from the SCN with two obligate relays, in the SPZ and DMH, for controlling the circadian rhythm of Cort secretion. Nearly all SPZ neurons express Vgat (are GABAergic), and the ablation of SPZ$^{Vgat}$ neurons eliminates the circadian rhythms of Cort secretion. However, deleting the *Vgat* gene from the SPZ has little effect on the Cort rhythm, suggesting that the message from the SPZ to the DMH cannot be conveyed by GABA alone. Ablation, deletion or inhibition of either DMH$^{Vglut2}$ or DMH$^{Vgat}$ neurons diminishes the circadian rhythm of Cort secretion, and the input from the DMH to the PVH$^{CRH}$ neurons takes the form of two parallel circuits with direct input from DMH$^{Vglut2}$ neurons exciting PVH$^{CRH}$ cells (the DMH$^{Vglut2}$ → PVH$^{CRH}$ pathway). By contrast, rDMH$^{Vgat}$ neurons disinhibit PVH$^{CRH}$ cells by rDMH$^{Vgat}$ neurons inhibiting intermediary cvPVH$^{Vgat}$ neurons (the rDMH$^{Vgat}$ → cvPVH$^{Vgat}$ → PVH$^{CRH}$ pathway). These circuits participate in the circadian rhythms of Cort secretion in a complementary way. The direct input from the DMH$^{Vglut2}$ neurons to the PVH$^{CRH}$ cells appears to be necessary to maintain the daily peak in Cort under both LD and DD, whereas the rDMH$^{Vgat}$ disinhibitory circuitry appears to be necessary for the daily increase in Cort levels at CT13 only during constant dark conditions. This suggests that the rDMH$^{Vgat}$ neurons mediate the disinhibition of PVH$^{CRH}$ neurons in anticipation of the circadian active period, but that under LD conditions the daily light signal itself can activate circuitry to disinhibit PVH$^{CRH}$ neurons at ZT13 even in the absence of the DMH$^{Vgat}$ circadian input. This relationship is similar to the "masking" phenomenon by which there is a reduction in LMA during the light phase of LD, even in mice in which the SCN has been ablated and which have no circadian rhythm of LMA in DD[24–26]. The origin of this light-induced signal for elevation of Cort at the beginning of the active period is not known, but apparently does not involve the DMH$^{Vgat}$ neurons.

This finding points out a *limitation in our study*, that although we specifically focused on the role of DMH neurons in the circadian regulation of Cort, DMH neurons may also participate in other Cort responses. For example, the DMH participates in stress responses, including the elevation of ACTH and Cort during stress[13,14,27–29]. Our work was designed to minimize stress during our circadian sampling by taking all blood samples within 1 min of touching the animal, so that we could isolate circuitry that contributes to the circadian regulation of Cort secretion in anticipation of the active period. As a control for whether this circadian circuitry also mediated stress-induced Cort responses, we also examined the Cort response to acute restraint in the same animals, and found that ablating either the glutamatergic or GABAergic neurons in the DMH had little if any effect on restraint stress-induced Cort levels. In contrast, the mice with ablation of cvPVH$^{Vgat}$ neurons showed an exacerbated Cort response to restraint, suggesting that the cvPVH$^{Vgat}$ neurons may tonically inhibit PVH$^{CRH}$ cells, even during restraint stress.

A second limitation in our study was that our ablations and deletions of SPZ$^{Vgat}$ neurons encroached upon the dorsal edge of the SCN, which is also composed of GABAergic neurons. Although previous studies have found that partial ablations or Vgat gene deletions of the SCN ablation do not abolish circadian rhythms of either Tb or LMA[30], it is possible that loss of 10–15% of SCN neurons may have

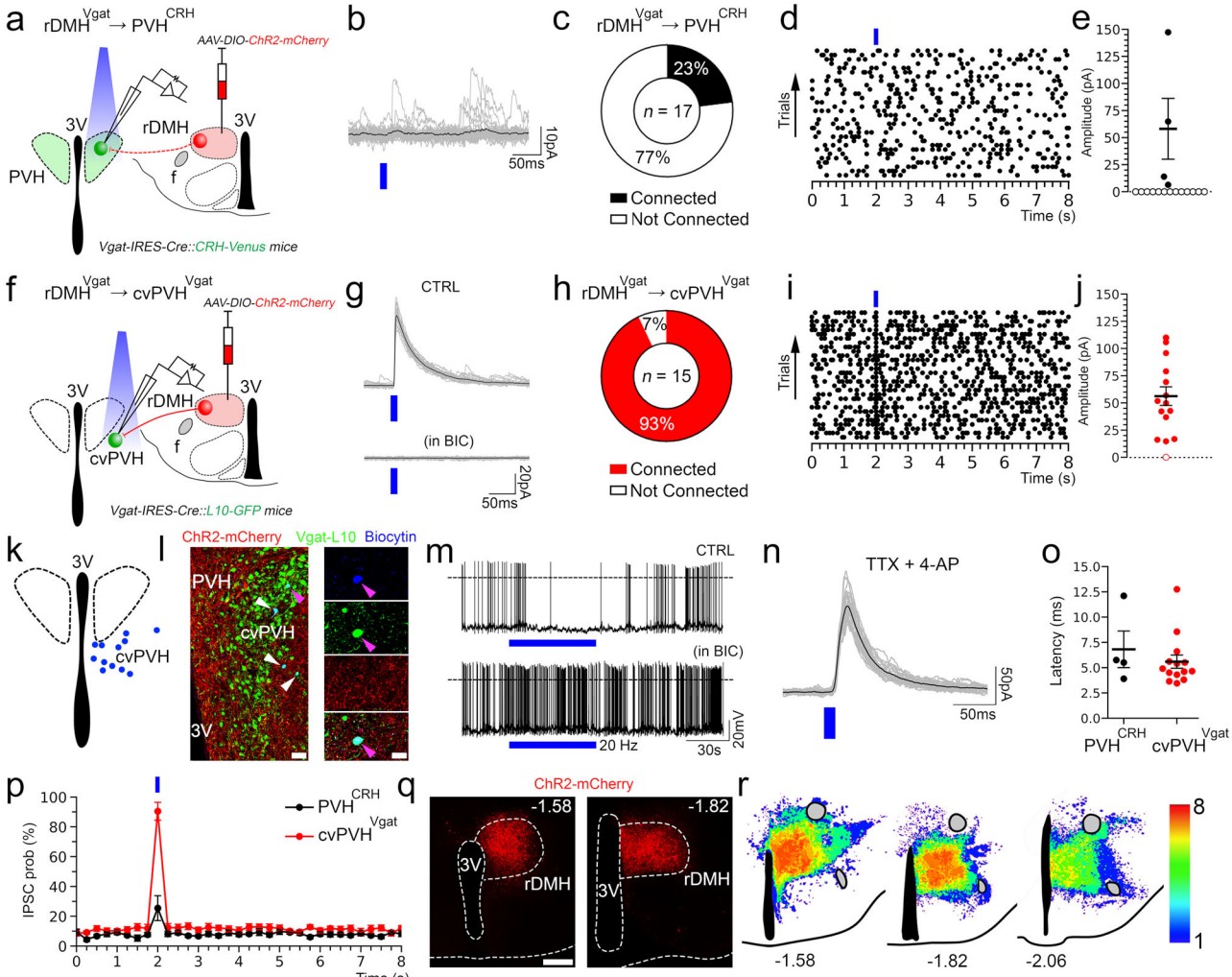

**Fig. 7 | In vitro optogenetic stimulation of the GABAergic input from the rostral DMH spares PVH[CRH] neurons and directly inhibits GABAergic neurons in the cvPVH. a** A schematic representation of the recordings to test rDMH[Vgat] → PVH[CRH] connectivity; *Vgat-ires-Cre::CRH-Venus* mice were injected with *AAV-DIO-ChR2-mCherry* in the rDMH ($n = 3$ mice). Recordings were conducted in brain slices from ipsilateral Venus-labeled PVH[CRH] neurons while photostimulating the ipsilateral DMH[Vgat] input (DMH is shown on the opposite side for ease of illustration). **b** Stimulation of the rDMH[Vgat] input produced no synaptic responses time-locked to the light pulses in most of the PVH[CRH] neurons. **c** Percentages of PVH[CRH] neurons responding (Connected) and not responding (Not Connected) to photo-stimulation of the rDMH[Vgat] input ($n = 17$ PVH[CRH] recorded neurons from 3 mice). **d** Raster plot of IPSCs in a representative PVH[CRH] neuron with photostimulation of the rDMH[Vgat] → PVH[CRH] input (bin duration: 50 ms) showing lack of input. **e** oIPSC amplitude following photostimulation of the rDMH[Vgat] → PVH[CRH] input (filled markers, cells responding to photostimulation, $n = 4$ neurons; open markers, cells not responding to photostimulation, $n = 13$; mean and ± SEM of responding neurons from 3 mice). **f** To explore rDMH[Vgat] → cvPVH[Vgat] connectivity, *Vgat-ires-Cre::L10-GFP* mice ($n = 5$ mice) were injected with *AAV-DIO-ChR2-mCherry* in the rDMH, and recordings were conducted in ipsilateral GFP-labeled cvPVH[Vgat] neurons while photostimulating the rDMH[Vgat] input. **g** Photostimulation of the DMH[Vgat] input produced opto-evoked inhibitory postsynaptic currents (oIPSCs) in most of the cvPVH[Vgat] neurons. GABA_A-mediated oIPSCs recorded in cvPVH[Vgat] neurons (*upper trace)* and blocked by bicuculline (BIC 20 μM; $n = 4$ neurons from 4 mice). **h** Percentages of cvPVH[Vgat] neurons responding (Connected) and not responding (Not Connected) to photostimulation of the rDMH[Vgat] input ($n = 15$ cvPVH[Vgat] recorded neurons from 5 mice). **i** Raster plot of IPSCs in a representative cvPVH[Vgat] neuron with photostimulation of the rDMH[Vgat] → cvPVH[Vgat] input (bin duration:

50 ms). **j** oIPSC amplitude following photostimulation of the rDMH[Vgat] → cvPVH[Vgat] input (filled markers, cells responding to photostimulation, $n = 14$ neurons; open markers, cells not responding to photostimulation, $n = 1$; mean and ± SEM of responding neurons from 5 mice). **k** A schematic map of 12 cvPVH[Vgat] neurons ($n = 5$ mice) that were recorded and found to receive input from DMH[Vgat] neurons. **l** A photomicrograph showing four recorded GABAergic cvPVH neurons that responded to photostimulation of DMH[Vgat] input (filled with biocytin from the recording pipette, indicated by arrowheads). DMH[Vgat] fibers expressing ChR2-mCherry (in red) surrounded the cvPVH[Vgat] neurons expressing GFP (in green). The neuron indicated by the magenta arrowhead is shown at higher magnification at the right, showing labeling, from top to bottom, for biocytin, GFP, mCherry, and merged. **m** Photostimulation trains (20 Hz, train frequency; 60 s, train duration and 10 ms, pulse duration) inhibited the activity of the cvPVH[Vgat] neurons ($n = 6$ neurons from 2 mice; *top*) and this effect was blocked by bicuculline (20 μM; $n = 2$ from 2 mice; *bottom*). **n** oIPSCs in cvPVH[Vgat] neurons recorded in the presence of TTX 1 μM + 4-AP 200 μM ($n = 6$ neurons from 1 mouse) indicating monosynaptic connectivity. **o** oIPSC latency recorded in PVH[CRH] (black; $n = 4$ neurons from 3 mice, mean and ± SEM) and cvPVH[Vgat] neurons (red; $n = 14$ neurons from 5 mice) following photo-stimulation of the DMH[Vgat] input. **p** oIPSC probability in PVH[CRH] (black; $n = 4$ neurons from 3 mice, mean and ± SEM) and cvPVH[Vgat] neurons (red; $n = 14$ neurons from 5 mice, mean and ± SEM) following photostimulation of the DMH[Vgat] input. **q–r** Photomicrograph of a representative rDMH injection with *AAV-DIO-ChR2-mCherry* (native signal) and density plots of injections in the rDMH ($n = 8$ mice, including experiments in (**a–e**) and (**f–n**) following which only 23% of PVH[CRH] neurons showed oIPSCs. Reference scale bar: in **l** (*left*) = 50 μm and (*right*) = 20 μm; in **q** = 250 μm. Atlas levels correspond to Paxinos & Franklin Atlas[52].

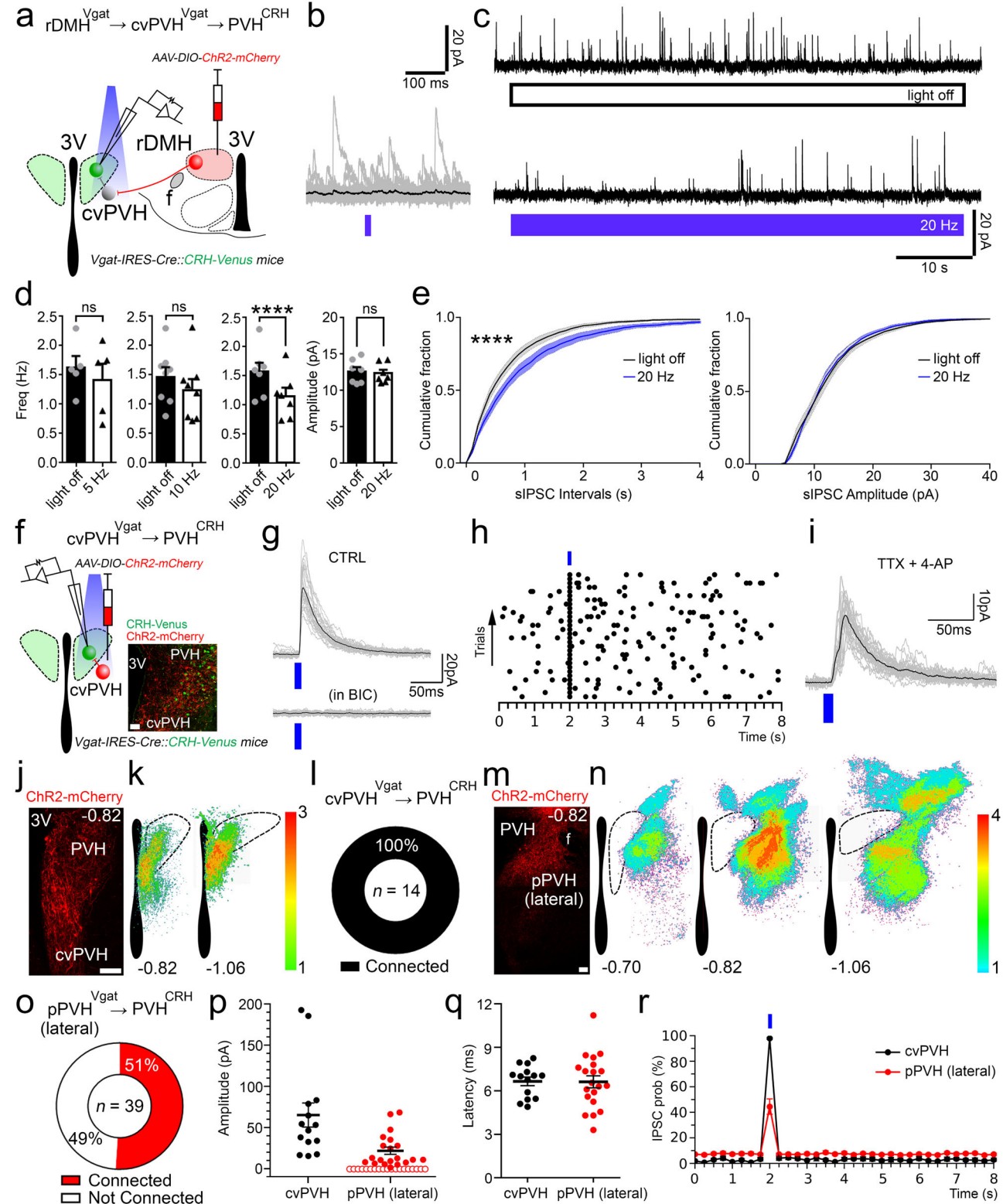

contributed to the reduction of LMA and Tb in our SPZ ablations or deletions. However, the reduction in the circadian rhythm in DD of Cort (about 98%), LMA (about 80% in DD) and Tb (about 60%) is more proportional to the approximately 75% cell loss in the SPZ.

Another limitation of our study was that while we examined the sources of circadian input from the DMH to the PVH^CRH neurons, our output measure was serum Cort levels. While CRH secretion is the dominant influence on secretion of ACTH by the pituitary gland, other hormones, such as arginine vasopressin may be secreted into the hypophysial portal circulation and can contribute to ACTH secretion[31]. However, *Crh−/−* mice, despite having normal circadian rhythms of LMA, lack (males) or have minimal (females) circadian increases in Cort secretion[32]. Thus, the focus on the PVH^CRH neurons as the main driver of Cort rhythms is justified.

A fourth limitation is that, although the DMH^Vgat neurons appear to be divided into a caudal DMH^Vgat group that directly inhibits PVH^CRH cells and a rostral DMH^Vgat group that disinhibits PVH^CRH neurons (by inhibiting cvPVH GABA neurons that innervate the PVH^CRH cells), our

**Fig. 8 | GABAergic neurons in the rostral DMH disinhibit the PVH$^{CRH}$ neurons via GABAergic neurons in the cvPVH. a** To test the rDMH$^{Vgat}$ → cvPVH$^{Vgat}$ → PVH$^{CRH}$ circuit, we injected *AAV-DIO-ChR2-mCherry* into the rDMH of *Vgat-ires-Cre::CRH-Venus* mice (*n* = 3 mice). Recordings were conducted in ipsilateral PVH$^{CRH}$ neurons while photostimulating rDMH$^{Vgat}$ synaptic terminals (DMH is shown on the opposite side of the brain for ease of illustration). We tested the effects of single light pulses (10 ms) and trains of photostimulation (10 ms individual pulse duration; 5, 10, and 20 Hz stimulation frequency; 60 s train duration) on spontaneous IPSC (sIPSC) frequency in PVH$^{CRH}$ neurons. **b** Single light pulses did not produce oIPSCs in most PVH$^{CRH}$ neurons, but (**c**) trains of stimulation reduced the sIPSCs frequency (*C; top* trace: *light off*; *bottom* trace: *light on*). **d** Change in sIPSC frequency following trains of photostimulation at 5, 10 and 20 Hz compared to sham stimulation (light off). Photostimulation at 5 and 10 Hz showed a trend toward reduction in sIPSC frequency that did not reach statistical significance (at 5 Hz: *n* = 5 neurons, *paired t-test, p* = 0.2772; at 10 Hz: *n* = 8 neurons, *paired t-test, Two-tailed, p* = 0.0545; from 3 mice, mean and ± SEM) whereas photostimulation at 20 Hz significantly reduced sIPSCs frequency (*n* = 7 neurons from 3 mice, *paired t-test, ****p* < 0.0001), without affecting their amplitude (*n* = 7 neurons from 3 mice; paired t-test, *p* = 0.7593). **e** Mean cumulative distribution plots of the sIPSC inter-event intervals show that intervals between sIPSCs were longer (left; 0.1 s bins; Kolmogorov-Smirnov test, two-sided, ****p* < 0.0001; 20 Hz vs light off) but that there was no change in sIPSC amplitude (right; 1pA bins; Kolmogorov-Smirnov test, two-sided, *p* = 0.1414; 20 Hz vs light off) as compiled from 7 PVH$^{CRH}$ neurons from 3 mice (blue: 20 Hz; black: light off; shaded areas: ± SEM). **f** To test the cvPVH$^{Vgat}$ → PVH$^{CRH}$ input, we injected AAV-DIO-ChR2-mCherry into the cvPVH of Vgat-ires-Cre::CRH-Venus mice (n = 3

mice). Recordings were conducted from ipsilateral PVH$^{CRH}$ neurons while photostimulating cvPVH$^{Vgat}$ neurons and terminals. **g** Stimulation of the cvPVH$^{Vgat}$ input evoked GABA$_A$-mediated oIPSCs in PVH$^{CRH}$ neurons (Bicuculline, 20 μM; *n* = 4 neurons from 3 mice). **h** Raster plot of IPSCs in a representative PVH$^{CRH}$ neuron showed tight correlation with photostimulation of the cvPVH$^{Vgat}$ → PVH$^{CRH}$ input (bin duration: 50 ms). **i** TTX -resistant oIPSCs in PVH$^{CRH}$ neurons (*n* = 6 neurons from 2 mice; TTX 1 μM + 4-AP 250 μM) indicating monosynaptic connectivity of the cvPVH$^{Vgat}$ → PVH$^{CRH}$ input. **j, k** A representative photomicrograph and density plots of injections of AAV-DIO-ChR2-mCherry in the cvPVH (**k**; *n* = 3 mice) and (**l**) percentages of PVH$^{CRH}$ neurons responding to photostimulation of the cvPVH$^{Vgat}$ input (Connected; n = 14 PVH$^{CRH}$ recorded *n*eurons from 3 mice). **m, n** Representative injection site and density plots of injections of AAV-DIO-ChR2-mCherry along the lateral margin of the PVH (pPVH) (n = 4 mice) and (**o**) percentages of PVH$^{CRH}$ neurons responding (Connected) and not responding (Not Connected) to photostimulation of the GABAergic input from the lateral pPVH (*n* = 39 neurons from 4 mice). **p** Amplitude (filled markers, cells responding to photostimulation, open markers, cells not responding to photostimulation; mean and ± SEM of responding neurons) and (**q**) latency of oIPSCs in PVH$^{CRH}$ neurons evoked by photostimulation of the cvPVH (black; *n* = 14 neurons from 3 mice) and lateral PVH (red; *n* = 39 neurons from 4 mice, mean and ± SEM). **r** oIPSC probability in PVH$^{CRH}$ neurons following photostimulation of the input from the cvPVH$^{Vgat}$ (black; *n* = 14 neurons from 4 mice and ± SEM) and lateral pPVH$^{Vgat}$ neurons (red; *n* = 39 mice 4 mice, mean and ± SEM). In all experiments we visualize only the native signal. Reference scale bar: in **f** = 50 μm; **j, m** = 100 μm. **f**, fornix, 3 V, third ventricle. Atlas levels are from Paxinos & Franklin Atlas[52]. ****p* < 0.0001.

---

stimulation, ablation, deletion, and inhibition studies probably included both groups. Because the caudal DMH$^{Vgat}$ neurons directly inhibit PVH$^{CRH}$ cells, any participation by these neurons in circadian regulation of Cort should be by suppressing secretion, e.g., during the daily nadir. However, DMH$^{Vgat}$ ablations or deletions had no effect on the daily nadir, and only affected the daily peak. The rostral DMH$^{Vgat}$ neurons, by contrast, disinhibit PVH$^{CRH}$ cells, and their ablation or deletion reduces the daily circadian peak of Cort secretion in DD. Interestingly, although the levels of Cort after chemogenetically activating the DMH$^{Vglut2}$ neurons were in the range typically seen during stress responses (>150 ng/ml), the peak levels seen after chemogenetic activation of the DMH$^{Vgat}$ neurons were <40 ng/ml. This lower level of Cort secretion may reflect the fact that both rDMH$^{Vgat}$ neurons which disinhibit PVH$^{CRH}$ neurons and caudal DMH$^{Vgat}$ neurons that inhibit CRH cells were being activated at the same time. Single nucleus RNA sequencing studies indicate that the DMH comprises more than 40 different cell types[33,34], roughly evenly divided between GABAergic and glutamatergic neurons. Further progress will require identifying the unique genetic signatures for the different classes of DMH neurons to test their roles in both circadian and stress-induced secretion of Cort.

### Relative contribution of DMH$^{Vglut2}$ and DMH$^{Vgat}$ neurons to circadian rhythms

It is also important to point out that the roles of the DMH$^{Vgat}$ and DMH$^{Vglut2}$ neurons in the circadian regulation of Cort secretion appear to depend upon release of GABA and glutamate, respectively, because deletion of just the *Vgat* or *Vglut2* genes from neurons in the DMH had effects similar to abating the entire Vgat or Vglut2 neuron population. The ablation of the SPZ$^{Vgat}$ neurons also caused almost complete loss of the circadian rhythm of Cort secretion and reduced the amplitude of the LMA rhythm by about 80% and reduced the Tb rhythm by about 60%. These effects on LMA and Tb were similar to a previous report using nonspecific SPZ ablation in rats[11]. However, the deletion of the *Vgat* gene in the SPZ had almost no effect on the circadian rhythm of Cort and only modestly reduced the of the amplitudes of LMA and Tb by about half as much as Vgat cell ablation. This disparity suggests that SPZ$^{Vgat}$ neurons cannot rely upon GABA alone to regulate these circadian rhythms. Information about the expression of peptides or other transmitters by SPZ neurons is limited so far, and will likely depend upon single cell mRNA expression profiling. On the other hand, many

DMH neurons also express peptides, including galanin, dynorphin, neuropeptide Y, CART, orexin, and many others[33–39]. While the different patterns of gene expression by DMH neurons may help us to classify them and provide key marker genes for neurons that participate in particular roles or have specific projections, it does not appear that the DMH neurons depend upon peptidergic transmission to produce the circadian pattern of Cort secretion.

In this work, we also were able to dissect the role of the DMH$^{Vglut2}$ vs DMH$^{Vgat}$ neurons in the global reduction of LMA and Tb and dramatic reduction in the circadian rhythm of LMA reported after non-specific ablation of the DMH in rats[12]. The DMH$^{Vgat}$ neurons appear to be particularly important for the circadian increase of LMA during the active period, as their ablation reduced the amount of LMA by about half across the entire dark or presumptive dark cycle. By contrast, ablation of the DMH$^{Vglut2}$ neurons caused a much smaller decrease in LMA which was predominantly confined to the transitions between the light and dark periods. The combination would explain the low levels of LMA with almost complete loss of circadian rhythms of LMA reported in rats after non-specific DMH cell ablation[12]. The ablation of either the DMH$^{Vglut2}$ or DMH$^{Vgat}$ neurons reduced Tb mainly during the dark and presumptive dark phase when the animals are most active. This effect is consistent with the known increase in Tb that occurs during LMA. In addition, dorsal hypothalamic/DMH glutamatergic neurons also play a direct role in promoting thermogenesis by projections to the raphe pallidus[29,40].

### Synergistic regulation of Cort rhythms by the DMH Glutamatergic and GABAergic populations

Previous work in rats had shown that non-specific ablation of the DMH in rats prevented the daily increase of Cort levels associated with the active cycle, reducing Cort secretion to the level normally measured during the inactive period when the animals are mainly asleep[12]. Because the effect of the DMH lesions was to eliminate the daily peak in Cort levels without affecting basal (nadir) levels, we initially hypothesized that the circadian input from the DMH to the PVH$^{CRH}$ neurons was likely to be excitatory, i.e., glutamatergic. In this study we tested that hypothesis and observed that the DMH$^{Vglut2}$ neurons play an important role in the circadian elevation of Cort at CT13 but that ablation of the DMH$^{Vglut2}$ neurons did not completely eliminate the circadian surge in Cort. We therefore tested the role of DMH$^{Vgat}$ neurons and were

surprised to find that they also participate in the circadian elevation of Cort, particularly in DD, at CT13 suggesting that the DMH[Vgat] neurons would be likely to disinhibit PVH[CRH] cells at this time.

These observations caused us to examine the origins of the GABAergic inputs to PVH[CRH] neurons. The PVH contains very few GABAergic neurons within its borders, whereas the pPVH is rich in GABAergic neurons and it has been proposed that the pPVH[Vgat] neurons could provide tonic inhibition of the PVH[CRH] neurons and function as a "brake" upon their secretion of CRH[23]. On the other hand, other studies have found that some of the GABAergic inputs to PVH[CRH] neurons derive from neurons that are distant from the PVH. For example, a recent study from Douglass and colleagues reported that the elevation of Cort levels during fasting depends upon AgRP neurons in the arcuate nucleus inhibiting tonically active GABAergic afferents from the bed nucleus of the stria terminalis to the PVH[CRH] neurons, thus disinhibiting them[18]. In the case of circadian secretion of Cort, we found that the rDMH[Vgat] neurons project to cvPVH[Vgat] neurons, and that cvPVH[Vgat] neurons provided a particularly rich source of inhibitory input to the PVH[CRH] cells. Furthermore, activation of DMH[Vgat] terminals in the PVH disinhibited PVH[CRH] cells. Interestingly, the surge in Cort levels at CT13 was greater after the ablation of the cvPVH[Vgat] neurons than in control animals, suggesting that the tonic activity of cvPVH[Vgat] neurons places a brake on CRH secretion that is only partially lifted during the circadian daily peak. This interaction suggests that disinhibition of the PVH[CRH] neurons may be a common motif in their regulation.

Although the evidence is strong for DMH[Vglut2] neurons directly exciting PVH[CRH] cells and DMH[Vgat] neurons disinhibiting PVH[CRH] cells via their inputs to the cvPVH[Vgat] neurons, we cannot rule out other polysynaptic contributions driven by the DMH[Vglut2] and DMH[Vgat] neurons. In addition, both neuron types in the DMH may contribute to regulation of Cort under other physiological conditions. Exploring these alternatives will require resolving the DMH glutamatergic and GABAergic neurons into discrete genetically identified cell types and determining their connections and physiological roles.

### Origin of the circadian signal to the DMH[Vglut2] → PVH[CRH] and DMH[Vgat] → cvPVH[Vgat] → PVH[CRH] pathways

Although the activities of the DMH[Vglut2] and DMH[Vgat] circadian inputs to the PVH[CRH] neurons are temporally aligned in control of the circadian rhythm of Cort, neither is in phase with the transitions between the light and dark periods. In fact, in mice the activity of PVH[CRH] neurons peaks about 6 h and the Cort levels rise about 3 h prior to the onset of the active (dark or presumptive dark) cycle, and reach a peak shortly after the active cycle begins, only to fall back to low levels by halfway through the active cycle[41,42]. Similarly, in humans the cortisol levels begin to rise about halfway through the habitual sleep cycle and are about twice as high at 8am as they are at 4 pm[43]. As indicated by our rabies virus tracing experiments (Supplementary Fig. 6b–d) and previous observations by others[4,5,42], there are few if any direct projections from the SCN to the PVH[CRH] neurons. It has been suggested that the sparse SCN[VIP] inputs to regions of the PVH that are nearby the CRH neurons may influence their firing via volume transmission[42]. However, our data indicate that such inputs are insufficient to cause the daily circadian surge in Cort secretion in the absence of relays in the SPZ and DMH. The fact that the timing of the increase in Cort secretion is offset by about 6 h from the light-dark cycle suggests that there is further circuitry downstream of the SCN that converts its signal, which is tied to the light-dark cycle, into a timing signal that starts the increase in Cort several hours prior to the onset of the active period. In addition, in nocturnal (mice) and diurnal (humans) species the onset of Cort secretion bears the same time relationship to onset of the active phase, even though their relationships to the light cycle (and the peak activity of SCN neurons) are 180 degrees out of phase with each other.

Here, our data suggest that the circadian pattern of Cort secretion depends upon a multi-synaptic circuit from the SCN to the SPZ and then the DMH. The ventromedial SPZ receives a large portion of the SCN output and projects heavily to the DMH in both rats and mice[6,15,44,45]. Most SPZ neurons have activity patterns that are in anti-phase to the SCN (i.e., are more active during the dark period) and they are almost uniformly GABAergic[9,10], but our results point to a different mechanism than GABA release in relaying the circadian rhythm of Cort secretion. The SPZ also receives inputs from a wide range of other hypothalamic cell groups. Thus, it is possible that various configurations of polysynaptic connections in the SPZ might allow it to produce timing signals that are driven by the SCN but not in phase with it. Such a multi-synaptic pathway is consistent with the smooth and gradual increase in Cort secretion that occurs prior to the active phase, allowing the SPZ[Vgat] neurons to drive the firing of DMH[Vgat] and DMH[Vglut2] neurons that are responsible for the circadian rhythm of Cort secretion. Further study of the intrinsic circuitry of the SPZ and the dynamics of its interaction with the DMH will be important for understanding how the SCN signal is used to optimally time various physiological processes.

## Methods
### Animals
Because adult female mice lose their circadian rhythm of Tb every 4-5 days during estrus and we needed to record circadian rhythms at specific times across many days, we used only male adult mice, age 12–16 weeks old, in these experiments. The strains of the mice were *Vgat-ires-Cre*[46] (JAX: 016962), *Vglut2-ires-Cre*[46] (JAX: 016963), *CRH-ires-Cre*[47] (JAX: 012704), *CRH-VenusΔNeo*[48], *Vglut2[loxP/loxP]* (JAX:036439), *Vgat[loxP/loxP]*[49] (JAX: 012897) and *R26-loxSTOPlox-L10-GFP* mice[47]. Mice were individually housed under 12–12 h light/dark cycle unless the protocol specified otherwise. Room temperature was controlled in a range of $22 \pm 2 \,^{\circ}\mathrm{C}$ and free access to food and water was provided. All procedures were performed in accordance with the National Institutes of Health Guide for the Care and Use of Laboratory Animals, and formal approval of our protocols was obtained from the Institutional Animal Care and Use Committee at Beth Israel Deaconess Medical Center. All precautions were taken to minimize pain and discomfort in the mice.

### Viral Vectors used
*AAV10-hSyn-mCherry-DIO-DTA*, which conditionally expresses the subunit A of diphtheria toxin in a Cre-dependent fashion and the mCherry protein in non-Cre cells (acquired from Patrick M Fuller)[50,51] was injected ( ~ 24nL) in *Vgat-ires-Cre or Vglut2-ires-Cre* mice. *AAV8-eSYN-EGFP-T2A-iCre* containing the genes for Cre and EGFP under the neuronal eSYN promoter (VB1089, Vector Biolabs, PA, US; $8.6 \times 10^{11}$ viral genomes ml⁻¹) was bilaterally injected ( ~ 24nL) in *Vglut2[loxP/loxP]* and *Vgat[loxP/loxP]* mice. *AAV8-Ef1a-DIO-ChR2(H134R)-mCherry* injections (UNC, addgene #AV9080, $1.4 \times 10^{13}$ viral genomes ml⁻¹) were used for the neuroanatomical and CRACM experiments ( ~ 3–9 nL) in *Vgat-ires-Cre or Vglut2-ires-Cre* mice. *AAV10-DIO-hGlyR-mCherry* (acquired from Patrick M Fuller)[16,29,45] injections were made in the DMH ( ~ 45 nL) either in *Vglut2-ires-Cre* or *Vgat-ires-Cre* mice. *AAV10-EF1α-DIO-hM3Dq-mCherry* (UNC, addgene #50460 virus core; $2 \times 10^{13}$ viral genomes ml⁻¹) was injected ( ~ 30nL) either in *Vglut2-IRES-Cre* or *Vgat-IRES-Cre* mice. *AAV8-CAG-DIO-EGFP* ( ~ 24 nl) was injected as a control virus in specified experiments (UNC; addgene #59331 $7 \times 10^{12}$ viral genomes ml⁻¹). ~ 45 nL of *AAV8-Ef1a-DIO-TVA-mCherry* (UNC Vector Core; $1.13 \times 10^{12}$ viral genomes ml⁻¹) mixed 1:1 with *AAV8-CAG-DIO-rabiesG* (Stanford Vector Core; $3.4 \times 10^{12}$ viral genomes ml⁻¹) was injected unilaterally, and 21 days later, ~ 55nL of *EnvA-ΔG-rabies-EGFP* (Salk Viral Vector Core; $1.51 \times 10^{8}$ viral genomes ml⁻¹) were administrated in the PVH of *CRH-ires-Cre* mice. All the experiments and recordings started

four weeks after the injections to allow complete transfection with the AAVs.

## Surgeries

Mice were deeply anesthetized with a mixture of ketamine/xylazine (100/10 mg/kg, i.p.). Radiotelemetry sensors (TA-F10, DSI, US) for body temperature (Tb) and locomotor activity (LMA) were implanted in the peritoneal cavity with a small abdominal incision. The coordinates for the stereotactic microinjections of viral vectors in the brain, from bregma were: DMH, AP = −1.75 mm, ML = ± 0.25 mm, DV = −4.9 mm; PVH, AP = −0.8 mm, ML = + 0.25 mm, DV = −4.8 mm; cvPVH, AP = −0.8 mm, ML = +0.25 mm, DV = −4.85 mm[52]. Mice receive analgesic treatment with meloxicam or buprenorphine post-surgery. All procedures were performed under aseptic conditions.

## LMA and Tb continuous recordings

Average Tb and LMA were recorded every 5 min during all the protocols using the radiotelemetry DSI system. The signal from the telemetry probes, previously implanted, was received and converted with the PhysioTel HD and PhysioTel (DSI) hardware. The LMA and Tb data were analyzed using the software ClockLab Analysis version 6 (Acticmetrics).

## Plasma sampling and corticosterone immunoassay

Blood samples (~20 μl) were obtained in a microvette tube (CB300, Sarstedt, US) after a small incision in the tail in less than 1 min after onset of handling the mouse, centrifuged at 9,240 × g for 10 min, and the plasma was collected and stored at −40 °C. Samples were collected with a minimum interval of 30 h between samples. Under the DD protocol, the first sample was obtained on the third day after the lights turned off at CT13 and then every 30 h until we completed the 4 time points. An ELISA assay for corticosterone was performed as per the manufacturer instructions (ADI-901-097, Enzo life science, US) and the corticosterone signal read with a 405 nm filter (iMark, Bio-Rad, US). Each sample was read in duplicates.

## Restraint stress protocol

At the end of the LMA and Tb protocols, mice were physically restrained for 1 h starting at ZT3 in a plastic tube with small holes to allow normal respiration. Blood samples were obtained at the end of the stress protocol by a small incision in the tail. The mice were placed in their home cage after the protocol.

## Perfusion and Immunohistochemistry

At the end of each protocol, animals were deeply anesthetized with 7% chloral hydrate (0.015 ml/gr of body weight i.p.) and perfused transcardially with 30 ml of PBS followed by 30 ml of 10%-buffered formalin (Thermo Fisher Scientific, US). Brains were extracted and postfixed in 10%-buffered formalin overnight, cryoprotected in 30% sucrose solution, sectioned into 40 μm coronal sections (three series) and stored at 4 °C in PBS with sodium azide. Brain sections were washed three times (5 min each) and then blocked for 1 h with 3% normal horse serum (diluted in 0.4% Triton X-100 in PBS). Sections were incubated in primary antibody overnight in the same solution as blocking, at room temperature with continuous agitation. Primary antibody used were rabbit anti VIP (ImmunoStar; 1:1000, #20077), Chicken anti-GFP (Invitrogen; 1:5000, A10262), Rat anti-mCherry (Invitrogen; 1:4000, M11217), Mouse anti-NeuN (Millipore; 1:3000, MAB377). Then, the sections were washed six times (5 min each) in PBS, and incubated with secondary antibodies for 2 h in blocking solution at room temperature and continuous agitation. The secondary antibodies used were donkey anti rabbit antibody conjugated with CY5 fluorophore (Jackson Immunoresearch; 1:1000, # 711-175-152), Alexa fluor 488-conjugated Donkey anti-Chicken (Jackson; 1:200, AB2340375), Alexa fluor 555-conjugated Goat anti-Rat (Invitrogen; 1:200, A21434), Alexa fluor 488-conjugated Goat anti-Mouse (Jackson; 1:200, AB2338840). Then, sections were washed three times (10 min each) in PBS, mounted on electrostatically treated slides and coverslipped with VECTASHIELD Antifade Mounting Medium with DAPI (Vector Laboratories) for microscope visualization and image acquisition.

## RNA scope in situ hybridization

Two brains from the *EnvA-ΔG-rabies-EGFP* experiments were sectioned at 20 μm and tissue was mounted on glass slides in RNAase-free conditions. Slices were dried and in situ hybridization was performed using RNAScope multiplex fluorescent reagent kit V2 (catalog #323100, Advanced Cell Diagnostics, US). Slices were pretreated with hydrogen peroxide for 10 min at room temperature and subjected to target retrieval step for 5 min in a steamer (>99 °C), This was followed by dehydration in 90% alcohol and air-dried for 5 min. Sections were then treated with protease III at 40 °C for 30 min. Then, the slices were rinsed with sterile water and incubated for 2 hrs either with the *Vglut2* probe (RNAscope Probe- Mm-Slc17a6; catalog #319171, Advanced Cell Diagnostics, US) or Vgat Probe (RNAscope Probe- Mm-Slc32a1; catalog # 319191, Advanced Cell Diagnostics, US) at 40 °C for RNA hybridization. This is followed by incubation with amplification reagents, AMP1, AMP2 (30 min each) and AMP3 (15 min) at 40 °C. Sections were then incubated with HRP -C1 for 15 min and Cy5 fluorophore (catalog #NEL741001, PerkinElmer) for 30 min at 40 °C. Finally, HRP blocker was added to the sections for 15 min at 40 °C. After each step sections were washed with 1 x wash buffer provided in the kit. Slides were then dried and coverslipped with Vectashield antifade mounting medium (catalog #H-1400, Vector Laboratories) for microscope visualization and image acquisition.

## Image acquisition, gradient maps and cellular counting

Coronal sections were scanned at 20X magnification using an Olympus VS120 slide-scanning microscope or at 63X magnification using a confocal microscope (Leica Stellaris 5). Fluorescence images were analyzed using Olympus OlyVIA software (3.4.1) or Leica application suite (version 4.2.1). Photomicrographs were adjusted for brightness and contrast to compensate for over- or undersaturation due to differences in fluorescence brightness. The injection site of each animal was confirmed by the expression of mCherry or EGFP. Microphotographs were transformed to a gray scale and the background were subtracted. The images at the same anatomical level were superimposed with 30% transparency to build a gradient map using GIMP software (version 2.10.34). In the brain sections from the rabies experiment, we counted cell bodies in at least 3 levels for each nucleus (anterior, middle and posterior). In the sections used for in situ hybridization of rabies-infected cells, we counted cell bodies and doubly-stained cells in 6 levels of the DMH. Cell body counting of the rabies infected cells was conducted manually using the multipoint tool in FIJI (Image J software, version 1.54). Because the number of infected cells varied considerably from animal to animal and we were not trying to establish the absolute number of cells that project to the PVH^CRH neurons, we did not attempt to correct the numbers for cell size.

## ChR2-assisted circuit mapping (CRACM)

For electrophysiology experiments, *AAV8-DIO-ChR2-mCherry* (~3–9 nL) was injected into the DMH of *Vglut2-ires-Cre::CRH-Venus* mice (n = 4) or *Vgat-ires-Cre::CRH-Venus* mice (n = 3) or *Vgat-ires-Cre::L10-GFP mice* (n = 5) or into the cvPVH of *Vgat-ires-Cre::CRH-Venus* mice (n = 7) or into the cneurons neurons

DMH of *Vgat-ires-Cre::CRH-Venus* mice (n = 4). Mice with AAV injections placed outside of the target were excluded from the analysis. Four to six weeks following the AAV injections, mice were deeply anaesthetized with isoflurane (5% in oxygen) via inhalation and transcardially perfused with ice-cold ACSF (N-methyl-D-glucamine, NMDG-based solution described below). The mouse brains were then quickly

removed and sectioned coronally (250 μm-thickness) in ice-cold NMDG-based ACSF using a vibrating microtome (VT1200S, Leica). We first incubated the slices containing the PVH for 5 min at 37 °C, then transferred them into a holding chamber at 37 °C containing ACSF (Na-based solution) for 10 min. We let the brain slices gradually return to room temperature (~1 h) before starting recording. Brain slices were recorded in a recording chamber, where were submerged, and perfused with Na-based ACSF (described below, 1–1.5 ml/min). We recorded PVH$^{CRH}$ and cvPVH$^{Vgat}$ neurons identified by Venus or L10-GFP (green) fluorescence, respectively, using a combination of fluorescence and infrared differential interference contrast microscopy (IR-DIC). Both the Venus-expressing CRH neurons and the L10-GFP neurons were numerous and clear under the microscopy without any enhancement protocol. We used a fixed stage upright microscope (BX51WI, Olympus America) equipped with a Nomarski water immersion lens (Olympus 40X / 0.8 NAW) and IR-sensitive CCD camera (ORCA-ER, Hamamatsu) to acquire real-time images using Micro-Manager software. We recorded the neurons in whole-cell voltage clamp and current clamp configurations using a Multiclamp 700B amplifier (Molecular Devices), a Digidata 1322 A interface, and Clampex 9.0 software (Molecular Devices). Neurons showing a greater than 10% change in input resistance over the duration of the recording were excluded from the analysis. We photostimulated the DMH or cvPVH axons in the PVH using a full-field (~10 mW/mm$^2$, 1 mm beam width) 5 W LUXEON blue light-emitting diode (470 nm wavelength; #M470L2-C4; Thorlabs), coupled to the epifluorescence pathway of the microscope. We stimulated photo-evoked excitatory postsynaptic currents (oEPSCs) or inhibitory postsynaptic currents (oIPSCs) with 10 ms light pulses (0.1 Hz, for a minimum of 30 trials). In the DMH$^{Vgat}$ injected mice, we also tested the effects of photostimulation on both the action potential firing of cvPVH$^{Vgat}$ neurons and the spontaneous IPSC (sIPSCs) of PVH$^{CRH}$ neurons. For sIPSC recordings, we used train stimulations with a duration of 60 s at frequencies of 5, 10, and 20 Hz, and a light pulse duration of 10 ms. For action potential recordings, we tested train stimulations of 10 and 60 s at the same frequencies (5, 10, and 20 Hz) with a light pulse duration of 10 ms. Action potential firing and oEPSCs (holding potential = −70 mV) were recorded in ACSF and using a K-gluconate-based pipette solution. oIPSCs (holding potential = 0 mV) were recorded in ACSF containing 1 mM kynurenic acid and using a Cs-methane-sulfonate-based pipette solution. To record photo-evoked synaptic events in the presence of synaptic blockade, we bath applied tetrodotoxin (TTX; 1 μM) and the potassium channel blocker, 4-AP (200–500 μM). For all recordings we added 0.5% biocytin in the pipette solutions to mark the recorded neurons. The recorded slices containing the PVH and cvPVH and the slices containing the DMH, where the AAVs were injected, were fixed overnight in 10%-buffered formalin for *post hoc* histological and anatomical assessment. To label and map the recorded neurons filled with biocytin, immediately after the in vitro recordings, we fixed the recorded slices in 10% buffered formalin, washed them, and incubated them overnight in streptavidin-conjugated Alexa Fluor 405 (1:500; Cat#: S32351; Invitrogen, Thermo Fisher Scientific Waltham, MA)[53,54]. We acquired images using a Leica Stellaris 5 confocal microscope using a 63X oil immersion objective.

## Solutions for CRACM experiments

NMDG-based ACSF solution containing (in mM): 100 NMDG, 2.5 KCl, 1.24 NaH$_2$PO$_4$, 30 NaHCO$_3$, 25 glucose, 20 HEPES, 2 thiourea, 5 Na-L-ascorbate, 3 Na-pyruvate, 0.5 CaCl$_2$, 10 MgSO$_4$ (pH 7.3: 95% O$_2$ and 5% CO$_2$; 310–320 mOsm). Na-based ACSF solution contained (in mM): 120 NaCl, 2.5 KCl, 1.3 MgCl$_2$, 10 glucose, 26 NaHCO$_3$, 1.24 NaH$_2$PO$_4$, 4 CaCl$_2$, 2 thiourea, 1 Na-L-ascorbate, 3 Na-pyruvate (pH 7.3–7.4 in 95% O$_2$ and 5% CO$_2$; 310–320 mOsm). Cs-methane-sulfonate-based pipette solution containing (in mM): 125 Cs-methane-sulfonate, 11 KCl, 10 HEPES,

0.1 CaCl$_2$, 1 EGTA, 5 Mg-ATP and 0.3 Na-GTP (pH adjusted to 7.2 with CsOH, 280 mOsm). K-gluconate-based pipette solution containing (in mM): 120 K-Gluconate, 10 KCl, 3 MgCl2, 10 HEPES, 2.5 K-ATP, 0.5 Na-GTP (pH 7.2 adjusted with KOH; 280 mOsm). We purchased tetrodotoxin and kynurenic acid from Cayman Chemical (Ann Arbor, MI) and bicuculline methiodide from Tocris Bioscience (Ellisville, MO). We purchased all other chemicals from Fisher Scientific (Waltham, MA) or Sigma-Aldrich (Saint Luis, MO).

## Data and statistical analysis for CRACM experiments

Recording data were analyzed using Clampfit 10 (Molecular Devices), MiniAnalysis 6 software (Synaptosoft), customized Python scripts (Python 3, www.python.org) and MatLab (version R2020B; MathWorks; Natick, MA) software. Figures were generated using Igor Pro version 6 (WaveMetrics), Prism 7 (GraphPad, La Jolla, CA), Inkscape (GitLab) and Photoshop (Adobe) software. To ensure unbiased detection of synaptic events, the EPSCs and IPSCs were detected and analyzed automatically using MiniAnalysis. We considered EPSCs or IPSCs in PVH$^{CRH}$ neurons and cvPVH$^{Vgat}$ neurons to be photo-evoked if their probability during the first 50 ms following the light pulses was greater than the IPSC probability + five times the standard error of the mean (SEM) before the light stimulation and if their latency was within ± 1 ms of the median value of E/IPSC latencies calculated for each neuron within the first 50 ms following photostimulation. We calculated the latency of the photo-evoked EPSC and IPSCs as the time difference between the start of the light pulse and the 5% rise point of the first synaptic event[54]. Group means were compared using *paired* t-tests. Cumulative distributions were compared using the *Kolmogorov-Smirnov* test. Values indicating $p < 0.05$ were considered significant.

## Statistical analysis

Data sets were tested to see whether they fulfilled the parametric criteria of Brown-Forsythe (homo/heteroscedasticity) and Shapiro-Wilk (fit to normal distribution). Corticosterone plasma levels and circadian distribution of LMA and Tb were analyzed with a *two-way ANOVA* with a factor for group and a factor for time, followed by a *post-hoc* multiple comparisons *Sidak* test. CI for Cort was calculated by the difference between the mean ZT13 and ZT1 levels and divided by the ZT13-ZT1 mean in each mouse, then normalized to 100% by the mean difference in the Control group. CI for LMA and Tb were calculated by the difference between the mean dark (or presumptive dark) and light (or presumptive light) levels divided by the 24 h total counts for LMA and 24 h mean for Tb, then normalized to 100% against the mean for the Control group. Light or Dark mean for LMA and Tb were evaluated with *two-way ANOVA* with a factor for group and a factor for LD vs DD, followed by a *post-hoc* multiple comparisons *Tukey* test. When the same mice were evaluated under different conditions as their own controls for LMA and Tb (pre- and post-DTA experiments, and the hGlyR experiments), we evaluate them with a *Repeated Measures [RM] two-way ANOVA* with a factor for group and a factor for LD vs DD, followed by a *post-hoc* multiple comparisons *Tukey* test. CI and amplitude data were compared by *unpaired* t-test analysis, or *one-way ANOVA*, followed by a *post-hoc* multiple comparisons *Tukey* test in the cases where included baseline in the analysis. Actograms, periodograms, and amplitude of the cosinor fitting were built using ClockLab ActiMetrics software version 6.1.02. Statistical analyses and graphs were performed with the GraphPad Prism software version 8. Data are represented as mean ± SEM. Significance values α were set at $p < 0.05$.

## Reporting summary

Further information on research design is available in the Nature Portfolio Reporting Summary linked to this article.

## Data availability

The data generated in this study are provided in the Source Data file. Source data are provided with this paper.

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

## Acknowledgements

This study was supported by NIH grants: R01 NS122589-03 and P01HL149630-04 to E.A. P01HL149630-04 and R01 NS085477 to C.B.S., and R03 NS128993-02 to RDL. The authors thank Quan Ha and Sathyajit Bandaru for the superb technical assistance.

## Author contributions

O.D.R-P., R.D.L., E.A., and C.B.S. designed research; O.D.R-P., R.D.L., N. L. S. M., D.E., M.A.K., S.S.B., N.V., and F.R. performed research; O.D.R-P., R.D.L., and E.A. analyzed data; C.B.S. contributed to the project administration and funding acquisition; O.D.R-P., R.D.L., E.A., and C.B.S. wrote the paper, and all the authors approved the final version. O.D.R-P., R.D.L., equally contributed to this study. E.A. and C.B.S. share senior authorship.

## Competing interests

The authors declare no competing interests.
