## [Transparent Peer Review file · Nature Communications]

A hypothalamic circuit for circadian regulation of corticosterone secretion

Corresponding Author: Dr Clifford Saper

Version 0:

Reviewer comments:

Reviewer #2

(Remarks to the Author)

In this manuscript the authors identify a multi-synaptic circuit controlling the daily release of corticosterone (CORT; SCN to SPZ to DMH to (cv)PVH and PVH). The authors' findings about the DMH to PVH circuit is very compelling: they show that DMH Vglut2 neurons directly activate PVH CRH neurons, and that DMH Vgat neurons disinhibit PVH CRH neurons by inhibiting GABAergic interneurons in the cvPVH. This dual-pathway organization is well supported by anatomical tracing, slice physiology, and bidirectional circuit manipulation, and represents an exciting finding that will be of interest to circadian biologists, endocrinologists, and neuroscientists. However, several key claims regarding the SPZ relay and the origin of circadian timing control are not supported by the data as currently presented. These issues substantially weaken the manuscript's central narrative.

1) Discussion Lines 591-594: "Previous studies [LeSauter and Silver 1999]... have shown that extensive loss of SCN neurons is required to cause loss of circadian rhythmicity. In our ablation experiments ... encroachment on the SCN was unlikely to cause the profound loss of CORT, LMA and Tb rhythms." The cited paper does indeed demonstrate that (LMA) arrhythmicity requires 75% of the SCN to be ablated, but this is not relevant to the authors' findings. The SPZ manipulations in this manuscript (neuron ablation and Vgat gene deletion), regardless of incidental SCN involvement, do not produce arrhythmicity in LMA or Tb (Ext. Figs. 1, 2). Even with SPZ ablation, LMA and Tb remain rhythmic by periodogram analysis. Thus, using an "abolition of rhythms" criterion to exclude SCN contributions to their observed amplitude attenuation is not justified.

2) Discussion Lines 595-598: "A recent study [Klett et al. 2024] using SCN Vgat deletion... found that unilateral or partial deletion in the SCN had little effect on LMA or Tb rhythms. Thus, involvement of the dorsal margin of the SCN ... was unlikely to affect those results." This reasoning is not supported. In the manuscript, the authors explicitly argue that Vgat neuron ablation and Vgat gene deletion produce different outcomes, and that this difference is due to mechanisms beyond GABA release. However, when discussing SCN involvement, the authors treat partial SCN Vgat gene deletion (in Klett et al.) as equivalent to partial SCN neuron ablation in their SCN ablation animals. This is logically inconsistent: if deletion and ablation are not interchangeable in the SPZ, they cannot be treated as interchangeable in the SCN. This citation therefore does not support the authors' conclusion that potential SCN damage in their ablation experiments is irrelevant.

3) Klett et al. 2024 also show that SPZ Vgat gene deletion using similar methods produces animals that maintain LMA and Tb rhythms in DD. In the current manuscript, the authors find that SPZ Vgat deletion results in an attenuation of these rhythms (to a lesser extent than SPZ Vgat neuron ablation, but an attenuation nonetheless). This raises an important interpretive issue the authors should address: either SPZ Vgat gene deletion is already known to be insufficient to abolish rhythms (as per Klett et al.) or the present study detects a real effect that differs from Klett et al. This discrepancy should be discussed.

4) Throughout the manuscript, "loss of rhythmicity" is used to describe the effects of their various manipulations on CORT, LMA, and Tb. However, the periodogram analyses presented in the supplemental figures show that LMA and Tb remain rhythmic. The authors' circadian index is primarily a measure of day-night contrast, not rhythmicity, per se. The analyses for CORT (two-way ANOVA) convincingly demonstrate arrhythmicity and/or loss/gain of rhythm amplitude, but categorical arrhythmicity for Tb and LMA is not actually demonstrated. This interpretation should be revised to reflect the attenuation, rather than elimination, of Tb and LMA.

5) Minor concern: the DTA validation data in Fig. 1b are difficult to interpret as presented. The representative image shows GABAergic neurons in green and mCherry expression in red, which the authors describe as "labeling non-GABA cells in the injection site." However, the relative intensity and extent of red mCherry expression appear markedly greater in the right SPZ compared to the left. If mCherry expression reflects cells lacking Cre (and thus not ablated by the DTA), one would expect comparable non-GABA labeling bilaterally following a bilateral injection. This asymmetry raises several questions: were the injections accidentally unilateral? Did viral spread differ between hemispheres? Or, is this example image an anatomical control (a purposeful unilateral injection) rather than a true experimental animal. Clarification of this point is needed to interpret the extent and specificity of DTA ablation.

Overall, the DMH to PVH to CORT findings are strong and exciting. Addressing these concerns would substantially strengthen the manuscript's central claims and better align the interpretation with the data as they are presented.

Version 1:

Reviewer comments:

Reviewer #2

(Remarks to the Author)

I appreciate the authors' careful and thoughtful responses to my prior comments. The revisions substantially improve the manuscript, particularly with respect to clarifying amplitude versus rhythmicity, addressing the SCN encroachment arguments, and reconciling the Klett et al. findings with the present data. Overall, these changes strengthen the paper, and the DMH to PVH CRH neuron circuit findings remain compelling and exciting.

I would like to raise one remaining point for consideration, which primarily concerns wording and interpretation rather than additional experiments.

Across the abstract, results, and discussion, the manuscript repeatedly frames the difference between SPZ Vgat neuron ablation and SPZ Vgat gene deletion as evidence that SPZ Vgat neurons use other transmitters (or that the message is conveyed "by another co-transmitter" or "predominantly by some other transmitter"). While the new quantitative comparisons are very helpful, I do not think the current data uniquely support this mechanistic inference.

Specifically, the finding that SPZ Vgat neuron ablation produces a larger reduction in circadian amplitude than Vgat gene deletion demonstrates that loss of GABA release alone is not sufficient to account for the full SPZ ablation phenotype. However, this disparity does not, by itself, establish that non-GABA transmitters are the primary or dominant mechanism of SPZ signaling. Alternative explanations remain plausible, including incomplete elimination of GABAergic signaling at relevant SPZ outputs, loss of local circuit architecture or structural connectivity with ablation, or contributions from cotransmission that are not directly tested here.

For this reason, I would encourage the authors to soften the mechanistic language in the abstract, results, and discussion. Statements such as "suggesting that they predominantly use some other transmitter" or "the mechanism is mediated at least in part by transmitters other than GABA" could be rephrased to emphasize insufficiency of GABA alone, rather than positive identification of an alternative transmitter. For example, framing the result as indicating that "GABA release alone cannot account for the SPZ contribution to circadian regulation" would, in my view, more accurately reflect what is demonstrated by the data.

Addressing this wording would bring the interpretation fully into alignment with the otherwise strong and carefully executed experimental results.

Responses to the reviewer are in RED below

Reviewer #2 (Remarks to the Author):

In this manuscript the authors identify a multi-synaptic circuit controlling the daily release of corticosterone (CORT; SCN to SPZ to DMH to (cv)PVH and PVH). The authors' findings about the DMH to PVH circuit is very compelling: they show that DMH Vglut2 neurons directly activate PVH CRH neurons, and that DMH Vgat neurons disinhibit PVH CRH neurons by inhibiting GABAergic interneurons in the cvPVH. This dual-pathway organization is well supported by anatomical tracing, slice physiology, and bidirectional circuit manipulation, and represents an exciting finding that will be of interest to circadian biologists, endocrinologists, and neuroscientists. However, several key claims regarding the SPZ relay and the origin of circadian timing control are not supported by the data as currently presented. These issues substantially weaken the manuscript's central narrative.

1) Discussion Lines 591-594: "Previous studies [LeSauter and Silver 1999]... have shown that extensive loss of SCN neurons is required to cause loss of circadian rhythmicity. In our ablation experiments ... encroachment on the SCN was unlikely to cause the profound loss of CORT, LMA and Tb rhythms." The cited paper does indeed demonstrate that (LMA) arrhythmicity requires 75% of the SCN to be ablated, but this is not relevant to the authors' findings. The SPZ manipulations in this manuscript (neuron ablation and Vgat gene deletion), regardless of incidental SCN involvement, do not produce arrhythmicity in LMA or Tb (Ext. Figs. 1, 2). Even with SPZ ablation, LMA and Tb remain rhythmic by periodogram analysis. Thus, using an "abolition of rhythms" criterion to exclude SCN contributions to their observed amplitude attenuation is not justified.

Response: The reviewer makes a good point. Although the previous studies of partial lesions of the SCN cited by LeSauter and Silver consistently failed to report loss of LMA rhythms with lesions of less than 75% of the SCN, none of them quantified either the lesion extent or the degree of loss of LMA rigorously (and none measured Tb), so that it is possible that partial lesions may have caused partial reduction of the amplitude of LMA (and Tb). We have therefore revised the text to state that the reduction of the amplitude of the circadian rhythms of LMA (by about 80% in DD) and Tb (by about 60% in DD) is proportional to the reduction in SPZ neuron number (by 75%) but not to the reduction in SCN neurons (by 10-15%).

2) Discussion Lines 595-598: "A recent study [Klett et al. 2024] using SCN Vgat deletion... found that unilateral or partial deletion in the SCN had little effect on LMA or Tb rhythms. Thus, involvement of the dorsal margin of the SCN ... was unlikely to affect those results." This reasoning is not supported. In the manuscript, the authors explicitly argue that Vgat neuron ablation and Vgat gene deletion produce different outcomes, and that this difference is due to mechanisms beyond GABA release. However, when discussing SCN involvement, the authors treat partial SCN Vgat gene deletion (in Klett et al.) as equivalent to partial SCN neuron ablation in their SCN ablation animals. This is logically inconsistent: if deletion and ablation are not interchangeable in the SPZ, they cannot be treated as interchangeable in the SCN. This citation therefore does not support the authors' conclusion that potential SCN damage in their ablation experiments is irrelevant.

Response: The reviewer is correct that the Klett paper only examined the effects of deletion of the Vgat gene in the SCN vs SPZ on circadian rhythms (of LMA and Tb). However, it showed that the deletion of the Vgat gene alone in the SCN is sufficient to cause SCN neurons to lose their synchrony and animals to completely lose their circadian rhythms. Thus, the deletion of the Vgat gene in the SCN, unlike the SPZ, does cause loss of rhythms equivalent to ablating the SCN. Nevertheless, we recognize that this is a weak argument, for the reasons cited in Response 1 (i.e., the effects of partial Vgat deletion in the SCN were not quantified). Hence, we have revised the text to remove this argument.

3) Klett et al. 2024 also show that SPZ Vgat gene deletion using similar methods produces animals that maintain LMA and Tb rhythms in DD. In the current manuscript, the authors find that SPZ Vgat deletion results in an attenuation of these rhythms (to a lesser extent than SPZ Vgat neuron ablation, but an attenuation nonetheless). This raises an important interpretive issue the authors should address: either SPZ Vgat gene deletion is already known to be insufficient to abolish rhythms (as per Klett et al.) or the present study detects a real effect that differs from Klett et al. This discrepancy should be discussed.

Response: The Klett paper was begun by Dr. Gompf and Fuller when they were postdocs in the Saper lab around 2010, with his input. However, the project moved with Dr. Gompf around 2013 to the Allen lab at the Univ of Oregon, and then eventually to the Fuller lab at UC Davis, and much of the work in the final manuscript was done by Dr. Klett as part of his PhD dissertation in the Allen lab. As a result, although quantification of circadian amplitude has been standard in the Saper lab since the early 2000's, it was not done in the Klett paper. Instead, the first and last authors chose to state that the circadian rhythms "remained strongly rhythmic" after Vgat gene deletions in the SPZ. This is actually consistent with the current data, with much more extensive and more rigorously characterized lesion locations, where quantification showed an approximately 45% reduction in the amplitude of the circadian rhythm of LMA and 30% reduction in Tb after SPZ Vgat gene deletions. By contrast, similarly extensive and well characterized SPZ Vgat neuron ablations in the current paper showed an approximately 80% reduction in the circadian amplitude of LMA and 60% reduction in Tb rhythm amplitude, i.e. Vgat neuron ablations caused about twice the reduction in the amplitude of circadian rhythms of LMA and TB compared Vgat gene deletions. Thus, we stand by our interpretation that a large part of the circadian signal from the SPZ for LMA and Tb must be mediated by other transmitters in those neurons. The text has been revised to reflect that analysis.

Actual % reductions in amplitude of circadian rhythms of LMA and Tb

	LMA		Tb	
	CI	Cosinor	CI	Cosinor
Vgat neuron ablation	82%	77%	59%	60%
Vgat gene deletion	48%	45%	31%	30%

4) Throughout the manuscript, "loss of rhythmicity" is used to describe the effects of their various manipulations on CORT, LMA, and Tb. However, the periodogram analyses presented in the supplemental figures show that LMA and Tb remain rhythmic. The authors' circadian index is primarily a measure of day-night contrast, not rhythmicity, per se. The analyses for CORT (two-way ANOVA) convincingly demonstrate arrhythmicity and/or loss/gain of rhythm amplitude, but categorical

arrhythmicity for Tb and LMA is not actually demonstrated. This interpretation should be revised to reflect the attenuation, rather than elimination, of Tb and LMA.

Response: We agree with the Reviewer that reduction in the amplitude of circadian rhythms should not be characterized as “loss of rhythmicity.” In fact, we used this term only once in our entire paper (not “throughout the manuscript”), and that was in quoting LeSauter and Silver (see Response 1), which has now been removed. Instead, because circadian rhythms may be maintained by multiple parallel pathways, it is more accurate simply to calculate the percent of reduction in the amplitude of the rhythm caused by each intervention. Contrary to the statement of the Reviewer (above), we never, at any point in our paper, claimed to demonstrate “arrhythmicity” or “elimination” of the circadian rhythm for Tb or LMA with any of our interventions, because that never occurred. In fact, the only rhythm that we completely eliminated was that of Cort, when we ablated the SPZ Vgat neurons.

We did claim reductions in the amplitude of the circadian rhythms of Tb and LMA in various experiments, where we used four different approaches to quantify this amplitude. First, we graphed the actual behavior of the animals across the entire 24 hour period in both LD and DD (Ext Figs 1 and 2 for SPZ). This presents all of the information concerning the variations in Tb and LMA across the day. We then employed three commonly used reductionist methods to quantify the circadian rhythm. The first is a periodogram, which shows circadian period length, but also can be used to give an estimate of the amplitude of the rhythm. (Because the animals are not all at exactly 24 hr period lengths, the best estimate of the rhythmicity is given by the area under the curve for the entire peak centered at 24 hr, not the height reached by the peak.) Because this calculation can be confounded by variations in the period length, we did not try to quantify the amplitude of the rhythm loss in the periodograms, but merely presented the graphs so the reader could draw his or her own conclusions. We also used the circadian index, which as the referee indicates measures the amount of the behavior during the light vs. dark (or presumptive light and dark) phases. This is most useful for behaviors that have a sharp square wave-like difference during the two phases, as LMA usually does. We also used a cosinor analysis, which fits the behavior to a sine wave pattern, which is more accurate for behaviors like Tb that vary in a sine wave type of circadian rhythm across the day, but may not align precisely with the light-dark phase. We presented ALL of the data in ALL FOUR formats for every experiment.

We have gone back over the text to make clearer which indices we are referring to and avoid vague terms like “loss of rhythmicity”.

5) Minor concern: the DTA validation data in Fig. 1b are difficult to interpret as presented. The representative image shows GABAergic neurons in green and mCherry expression in red, which the authors describe as “labeling non-GABA cells in the injection site.” However, the relative intensity and extent of red mCherry expression appear markedly greater in the right SPZ compared to the left. If mCherry expression reflects cells lacking Cre (and thus not ablated by the DTA), one would expect comparable non-GABA labeling bilaterally following a bilateral injection. This asymmetry raises several questions: were the injections accidentally unilateral? Did viral spread differ between hemispheres? Or, is this example image an anatomical control (a purposeful unilateral injection) rather than a true experimental animal. Clarification of this point is needed to interpret the extent and specificity of DTA ablation.

Overall, the DMH to PVH to CORT findings are strong and exciting. Addressing these concerns would

substantially strengthen the manuscript's central claims and better align the interpretation with the data as they are presented.

Response: The reviewer is correct that the image in Fig. 1b is difficult to interpret. This is largely because it was taken with a slide scanner, which uses preset parameters, and does not adjust the image for oversaturation or undersaturation. However, in the original image, it is quite clear that there are substantial numbers of red-labeled cells on the left, they are just dimmer than the red cells on the right. (This happens when some cells are transduced by more than one viral particle; however, it takes only one viral particle to cause a recombination event in the cell, so the entire population of transduced cells is critical, not just the brightest ones). We have adjusted the red channel brightness and contrast on the two sides of the image to show more clearly the locations of the red cells on each side of the brain, which mark the location of the surviving cells that do not express Cre and were included within the injection sites. As with most if not all bilateral injection sites, they are not quite symmetric. The injection on the left was a bit more lateral than the one on the right. As a result, there are more residual SPZ-Vgat neurons on the left than the right. Our quantitative data on cell counts for the SPZ and SCN take this into account. With the editor's permission we would like to substitute the image below for the original one, as we think this more clearly shows the data. We have added into the text for the methods that images were corrected for brightness and contrast.

Response to Reviewer

Our responses are below in **RED**.

Reviewer #2 (Remarks to the Author):

I appreciate the authors' careful and thoughtful responses to my prior comments. The revisions substantially improve the manuscript, particularly with respect to clarifying amplitude versus rhythmicity, addressing the SCN encroachment arguments, and reconciling the Klett et al. findings with the present data. Overall, these changes strengthen the paper, and the DMH to PVH CRH neuron circuit findings remain compelling and exciting.

Thank you.

I would like to raise one remaining point for consideration, which primarily concerns wording and interpretation rather than additional experiments.

Across the abstract, results, and discussion, the manuscript repeatedly frames the difference between SPZ Vgat neuron ablation and SPZ Vgat gene deletion as evidence that SPZ Vgat neurons use other transmitters (or that the message is conveyed "by another co-transmitter" or "predominantly by some other transmitter"). While the new quantitative comparisons are very helpful, I do not think the current data uniquely support this mechanistic inference.

Specifically, the finding that SPZ Vgat neuron ablation produces a larger reduction in circadian amplitude than Vgat gene deletion demonstrates that loss of GABA release alone is not sufficient to account for the full SPZ ablation phenotype. However, this disparity does not, by itself, establish that non-GABA transmitters are the primary or dominant mechanism of SPZ signaling. Alternative explanations remain plausible, including incomplete elimination of GABAergic signaling at relevant SPZ outputs, loss of local circuit architecture or structural connectivity with ablation, or contributions from cotransmission that are not directly tested here.

For this reason, I would encourage the authors to soften the mechanistic language in the abstract, results, and discussion. Statements such as "suggesting that they predominantly use some other transmitter" or "the mechanism is mediated at least in part by transmitters other than GABA" could be rephrased to emphasize insufficiency of GABA alone, rather than positive identification of an alternative transmitter. For example, framing the result as indicating that "GABA release alone cannot account for the SPZ contribution to circadian regulation" would, in my view, more accurately reflect what is demonstrated by the data.

Addressing this wording would bring the interpretation fully into alignment with the otherwise strong and carefully executed experimental results.

Thank you, we have made this change in lines 62-63, 175-176, and 561 and 639.